# CORE: Conflict-Oriented Reasoning for General Multimodal Manipulation Detection

**Jinjie Shen** [1 2 3]  **Yaxiong Wang** [† 1 3]  **Yujiao Wu** [4]  **Lechao Cheng** [1 3]
**Tianrui Hui** [1 3]  **Nan Pu** [1 3]  **Zhihui Li** [5]  **Zhun Zhong** [† 1 3]

## Abstract

The rapid rise of generative AI has made multimodal fake news increasingly realistic and pervasive, posing severe threats to public trust and social stability. Existing detection methods rely heavily on manipulation-specific models and large-scale labeled data, resulting in poor generalization to emerging manipulation types. We observed that the essence of manipulated misinformation lies in its intrinsic conflicts, **i.e.,** semantic or physical inconsistencies either across modalities or with common world knowledge. Inspired by this observation, we propose **C**onflict-**O**riented **RE**asoning (**CORE**) framework, an effective paradigm that learns to endows multimodal large language models (MLLMs) with explicit conflict-capturing capability. To this end, CORE first constructs the Conflict Attribution Corpus (CAC) with fine-grained annotations of conflict factors and sources, providing essential data support for subsequent conflict perception training. By performing conflict-oriented representation enhancement and reasoning based on CAC, CORE achieves robust and generalizable conflict detection, effectively and rapidly adapting to unseen manipulation types with a few samples or in even zero-shot settings. Extensive experiments demonstrate that CORE surpasses state-of-the-art models. The dataset and code are publicly available at https://github.com/shen8424/CORE.

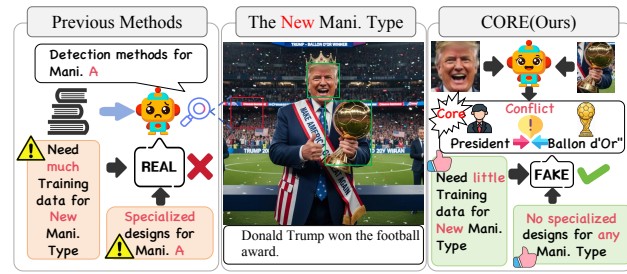

*Figure 1.* While previous methods require extensive data and specialized designs for specific manipulations, they struggle with new types. Our CORE addresses the core "conflict" in fake news, enabling generalized detection and excellent performance with minimal data. "Mani." means "Manipulation"

## 1. Introduction

The rapid advancement of generative artificial intelligence is profoundly impacting multiple domains (Haydarov et al., 2024b; Li et al., 2024; Abdelnabi et al., 2022; Jiang et al., 2020a), deeply blurring the boundary between reality and fiction. In social network, malicious actors can now create highly convincing multimodal fake news, combining manipulated images with deceptive text, at an unprecedented scale and speed (Yu et al., 2024; Haydarov et al., 2024a; Jiang et al., 2020b; Li et al., 2020a; Lu et al., 2023). These forgeries, ranging from subtle edits of facial attributes to entirely fabricated scenes, pose a serious threat to public trust, social stability (Lu et al., 2023; Li et al., 2020b; Shao et al., 2022). As manual verification becomes increasingly difficult, the development of robust automated detection systems is more critical than ever.

In response to these challenges, various manipulation detection methods have been developed (Shao et al., 2023; Shen et al., 2025; Zhang et al., 2025a; Liu et al., 2024; Shao et al., 2024; Bei et al., 2024), successfully alleviating the rampant spread of multimodal fake news. However, the success of these methods is predicated on designing models and training paradigms tailored to specific manipulation types and relying on large-scale, type-specific training data. In practice, there is a continuous "arms race" between forgery techniques (Chen et al., 2020; Wang et al., 2022a; Patashnik et al., 2021; Gao et al., 2021) and detection methods. The evolutionary pace of new manipulation methods far outpaces

---

[†]Corresponding author. [1]School of Computer Science and Information Engineering, Hefei University of Technology, Hefei, China [2]Wuhan University, Wuhan, China [3]Lab for Intelligence and visiON (LION) [4]CSIRO, Australia [5]University of Science and Technology of China, Hefei, China. First author: Jinjie Shen <shenjinjie22@gmail.com>. Corresponding to Yaxiong Wang <wangyx@hfut.edu.cn>, Zhun Zhong <zhunzhong007@gmail.com>.

*Proceedings of the 43rd International Conference on Machine Learning*, Seoul, South Korea. PMLR 306, 2026. Copyright 2026 by the author(s).

both the cycle of data collection, cleaning, and annotation, and the targeted model design required for each novel type. As a result, current methods significantly degrades when encountering new manipulation patterns (Zhang et al., 2025c; 2026; Lian et al., 2026). Therefore, the field urgently requires a new paradigm that can move beyond dependencies on data and specific model designs, enabling models to achieve effective adaption with only a few samples to novel manipulation types (Brown et al., 2020; Wang et al., 2022b; Madaan et al., 2022).

We observed that the essence of manipulated information lies in its intrinsic "conflict". This conflict can manifest as: a semantic contradiction between the **content** and **world knowledge**, such as the common-sense conflict between Trump's presidential status and football award in the news "Donald Trump wins the football award"; or a conflict at the physical level, such as lighting and shadows, between the manipulated content and the original image/text. As shown in Figure 1, existing methods implicitly capture such conflict with massive training data and specialized model designs, but this over-reliance leads to overfitting towards specific manipulation patterns, resulting in poor generalization for new types. In contrast, humans detect deception by activating their knowledge and performing conflict reasoning, enabling robust judgment across diverse manipulation forms. Motivated by this consideration, we argue that *if a detection model is endowed with explicit conflict-capturing capability, it can emulate human-like robustness when facing novel manipulation scenarios, thereby alleviating the long-standing data dependence and design rigidity of current approaches.*

Following the human reasoning process in detecting manipulations in multimodal misinformation, the ability to capture multimodal conflicts largely depends on a model's understanding of real-world knowledge. Multimodal Large Language Models (MLLMs) (Bai et al., 2025; Team, 2025b; 2024; Guo et al., 2025), trained on vast multimodal corpora, inherently encode rich world knowledge and thus exhibit strong potential for identifying conflicts in multimodal manipulations. However, they still fall short in conflict capturing due to the lack of conceptual understanding. MLLMs often conflate entirely unrelated concepts in the feature space, such as *"U.S. President"* and *"Football Award"* (Sec. 3 Figure 2a). Owing to this weakness, existing MLLMs, despite their rich world knowledge, still struggle to achieve robust and generalizable misinformation detection (Table 2).

To overcome the aforementioned limitations and establish a foundation model for general multimodal manipulation detection, we propose the Conflict-Oriented REasoning (CORE) framework. This framework equips MLLMs with explicit conceptual understanding, thereby enabling conflict detection capabilities. Training this capability requires explicit, fine-grained conflict supervision, which existing datasets lack. To provide this necessary data support, we first construct the Conflict Attribution Corpus (CAC). Each sample in CAC is annotated with both a conflict factor revealing the specific contradictory content within the misinformation and a conflict source, indicating whether the contradiction arises from the text, image, or underlying world knowledge. With these fine-grained annotations, we perform a Conflict-Perception Training (CPT) to perceive multimodal conflicts by enhancing the boundaries between conflicting concepts in the feature space, acquiring human-like conflict comprehension and detection ability.

With the acquired conflict-capturing capability from CPT, our CORE framework enables rapid adaptation to emerging manipulation patterns. Superior detection performance can be achieved with only a few-sample fine-tuning of new manipulation types, and even under zero-shot settings. In summary, our main contributions are as follows:

**(1)** We introduce an effective learning paradigm for general multimodal manipulation detection, which can rapidly adapt to novel manipulations with limited target samples.

**(2)** Moving beyond the conventional paradigm of designing models for specific manipulations, we propose CORE, a general framework for multimodal misinformation detection that endows MLLMs with human-like conflict reasoning and enables fast adaptation to unseen misinformation patterns.

**(3)** We construct the Conflict Attribution Corpus (CAC), a carefully curated dataset containing 19,532 samples with fine-grained annotations of conflict factors and sources, providing a solid benchmark for studying conflict reasoning in multimodal manipulation.

**Conflict of Interest Disclosure.** The authors declare no conflicts of interest.

## 2. Related Works

As deepfake technologies continue to evolve, research in the field of multimodal disinformation detection has also made significant progress. For instance, models such as HAMMER (Shao et al., 2023) and ASAP (Zhang et al., 2025b) have designed specialized contrastive learning and fine-grained detection modules to address the specific problem of image-text inconsistency; meanwhile, RamDG (Shen et al., 2025) focuses on celebrity-related fake news, employing external knowledge bases for targeted detection. In recent years, the rise of MLLMs has pushed research to new heights. SNIFFER (Qi et al., 2024) designed a specialized two-stage fine-tuning process to enhance the ability to judge image-text consistency, while FKA-Owl (Liu et al., 2024) attempts to tackle specific types of common sense fallacies by integrating world knowledge. To handle more complex forgeries, MMD-Agent (Liu et al., 2025) constructs a spe-

*Table 1.* World Knowledge Evaluation of Non-MLLMs and MLLMs.

*(a)* Average world knowledge evaluation, comparing Non-MLLMs (CLIP, ALBEF) against MLLMs (Qwen2.5VL-3B, Gemma3-4B).

| Models | World Knowledge (ACC %) |
|---|---|
| Non-MLLMs | 41 |
| MLLMs | **96** |

*(b)* Linear classification accuracy. Results are averaged across textual and visual modalities.

| Classification Task | ACC (%) |
|---|---|
| Pres. vs. Football Award | 61 |
| Pres. vs. UK Prime Minister | 53 |

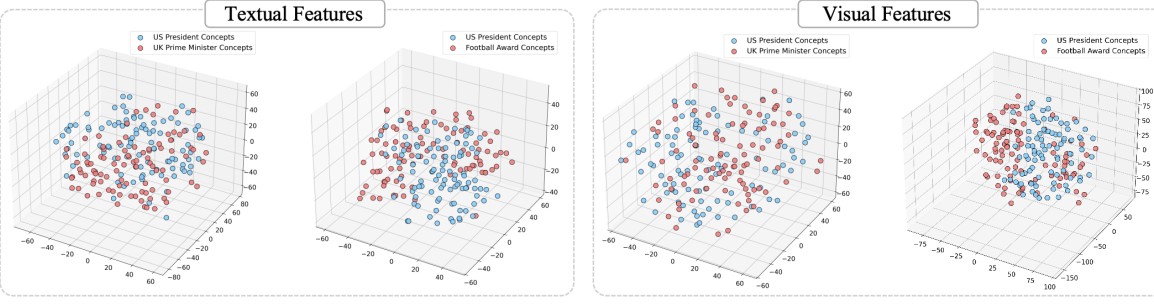

*(a)* Multimodal feature distribution of Qwen2.5VL-3B.

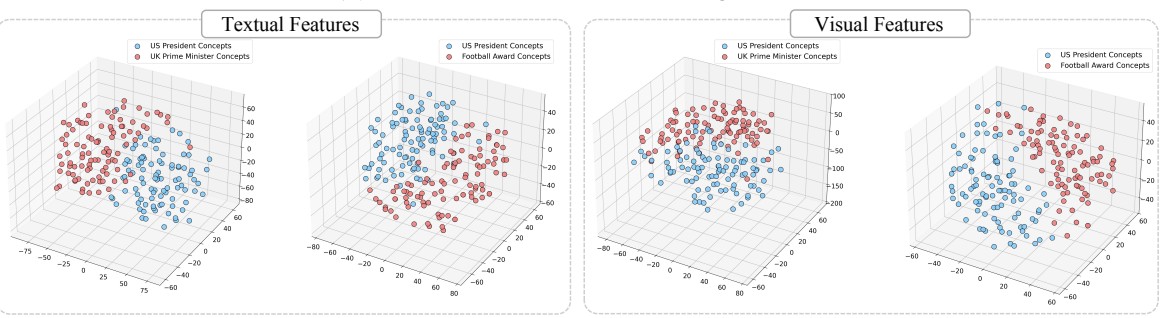

*(b)* Multimodal feature distribution of CORE$_{\text{Qwen}}$(2.5VL-3B).

*Figure 2.* Multimodal feature visualization of two group of conceptions from Qwen2.5VL-3B (a) and Qwen2.5VL-3B equipped with our CORE (b), where the textual and visual features are respectively shown from left to right.

cific multi-step reasoning framework, and AMD (Zhang et al., 2025a) relies on detailed prior information, such as manipulation region coordinates and manipulation types, for detection.

Despite the considerable progress in multimodal news detection methods, they suffer from two limitations. First, they heavily rely on large-scale datasets constructed for specific manipulation types; second, their model designs or training strategies are often specialized for certain forgery traces. These specialized designs make it difficult to guarantee the models generalization ability when faced with out-of-distribution, and especially unseen, manipulation types. Therefore, our work moves away from designing for specific forgery traces and instead focuses on the core flaw of forged information—conflict. Mastering this fundamental capability allows the model to break its dependence on large-scale, specific data, thereby demonstrating robust generalization and detection capabilities for unseen manipulation types in a few-sample and even zero-shot scenarios.

## 3. The Challenge of Conflict Perception

Human often do not require extensive training on similar samples to identify novel forged information. This is largely attributable to their ability to acutely identify conflicts within news based on their world knowledge and understanding. This capability is built upon two core foundations: 1) a comprehensive repository of world knowledge, and 2) a clear and well understanding of that knowledge to support conflict-capturing. This section investigates through a series of experiments whether current mainstream models possess these two key capabilities.

We first investigate a fundamental question: *Do existing models possess the world knowledge required to identify fake news?* To this end, we constructed a benchmark of 200 multiple-choice questions, covering the diverse world knowledge needed to detect fake news (See Appendix G for details). Our evaluation includes two representative classes of models: non-MLLM models, such as CLIP (Radford et al., 2021) and ALBEF (Li et al., 2021), and MLLMs,

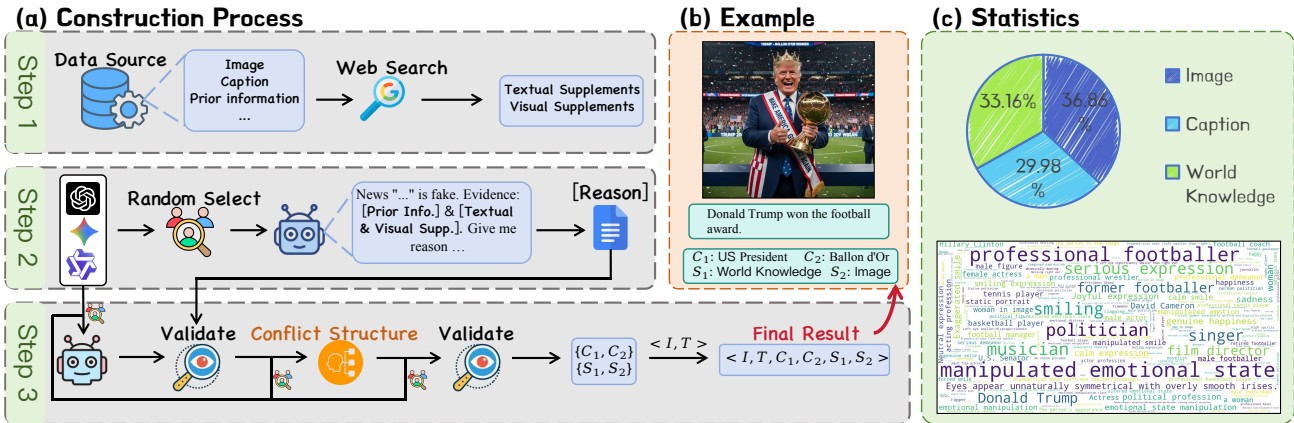

*Figure 3.* (a) The construction process of CAC. (b) An example from CAC. (c) Statistics of CAC, including the distribution of conflict sources and word clouds of the Conflict Factor.

including Qwen2.5VL-3B and Gemma3-4B. For the non-MLLM models, we assess their choices by calculating the cosine similarity between the embeddings of the question and the options after they pass through the encoder; the option with the highest similarity is considered the model's prediction. For MLLMs, we directly use prompting to have them output the correct option. The experimental results, as shown in Table 1a, indicate that MLLMs possess relatively complete knowledge, whereas non-MLLMs do not.

To investigate whether MLLMs possess clear conceptual boundaries like humans, we systematically analyze their feature representation space. We select concept pairs with varying semantic differences (*e.g.*, U.S. President vs. football player; U.S. President vs. UK Prime Minister), collect 100 relevant entities for each concept, and extract their multimodal features (See the Appendix F for details). We then use t-SNE (Van der Maaten & Hinton, 2008) for visualization. As shown in Figure 2a, the results demonstrate that the MLLM's representation space fails to form clear boundaries: the distributions for even semantically disparate concepts are diffuse and overlapping; We further train a classifier based on the features to quantify their separability, the low classification accuracy in Table 1b also quantitatively confirms this.

The experiments show that non-MLLM models suffer from incomplete knowledge, while MLLMs, though addressing the knowledge repository issue, still lack conceptual clarity. The key to resolving this dilemma is to enable models to possess knowledge and understand it in a clear, structured way, thereby learning detection based on the core principle of conflict.

## 4. Methodology

To overcome the above limitations, we propose the Conflict-Oriented Reasoning (CORE) framework, as illustrated in

Figure 4. Built upon MLLMs, CORE leverages their extensive knowledge base and reshapes the boundaries between conflicting concepts to enhance the model's conceptual understanding, thereby improving conflict perception capability and enabling MLLMs with human-like conflict capturing ability. To enable this, we first construct the Conflict Attribution Corpus (CAC) with fine-grained annotations of conflict sources and factors. Next, Modality Bridging Pre-Training (MBPT) is conducted to train a Cross-Modal Aligner. This aligner bridges the modality gap, enabling the full utilization of CAC annotations. Finally, the Conflict Perception Training (CPT) stage explicitly reshapes the model's conceptual understanding of conflicting elements, thereby refining its ability to perceive and reason over multimodal conflicts.

### 4.1. Conflict Attribution Corpus

The CAC provides explicit supervision signals to facilitate conflict perception learning. As illustrated in Figure 3(a), the construction of CAC involves the following steps:

**-Source Sample Selection.** Given that the SAMM (Shen et al., 2025) provides rich manipulation annotations including manipulated objects and regions, which offer valuable cues for subsequent conflict attribution generation. We therefore select 100k image-text pairs from it as our base data.

**-Background Knowledge Collection.** Next, we collect external supplementary materials that are semantically related to each image–text pair as background knowledge via the Google Search API (Google, 2025), providing reliable and comprehensive support for conflict reasoning.

**-Conflict Rationale Generation.** Subsequently, the integrated information including the multimodal inputs, manipulation prior and the background information are fed into a MLLM randomly selected from an expert pool of {GPT-4o, Gemini2.5-Pro, Qwen3-VL-Plus} (4o Team, 2024; Team,

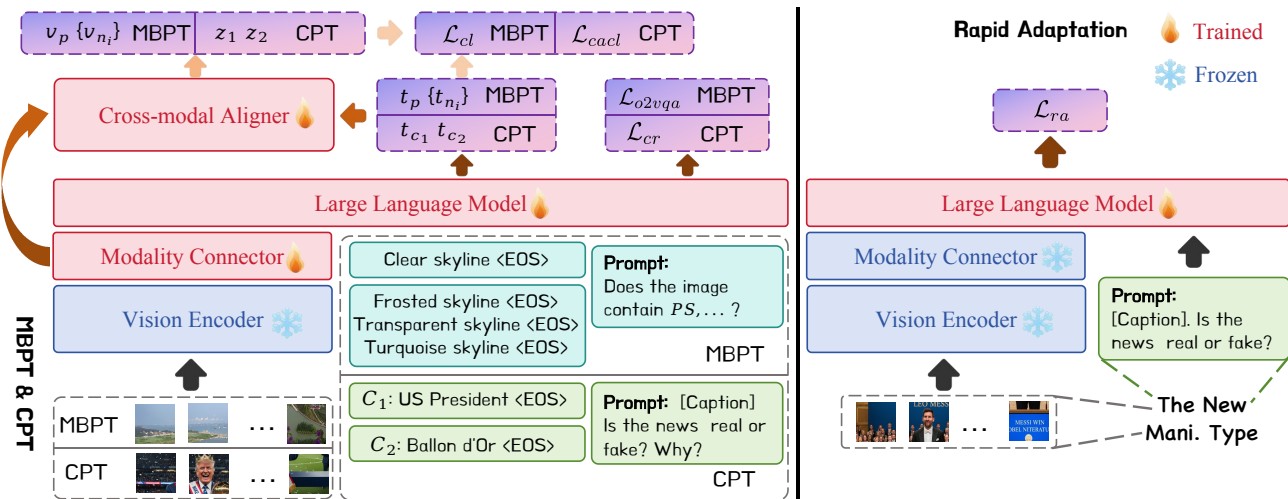

*Figure 4.* The architecture of our proposed CORE. It first employs MBPT to train cross-modal alignment, subsequently utilizes CPT to train conflict perception, and finally achieves effective detection of novel manipulations via Rapid Adaptation.

2025a; Bai et al., 2023), instructing it to generate a detailed reason why the news is false. The conflict rationale is then cross-validated for plausibility by the other two MLLMs.

**-Conflict Structuring**. After validation, the sample is once again sent to a randomly selected MLLM to distill the reason into a structured format of Conflict Factor 1, Conflict Factor 2 and their respective Conflict Source 1, Conflict Source 2, where the conflict factor specifies the content of the contradiction, while the conflict source pinpoints its origin, as shown in Figure 3(b). This result also undergos a final review by the remaining two MLLMs.

**Statistics.** As shown in Figure 3(c), CAC contains 19,532 instances. Its final data structure is <Image $I$, Text $T$, {Conflict Factor $C_1$, Conflict Factor $C_2$}, {Conflict Source $S_1$, Conflict Source $S_2$}>. Regarding the distribution of conflict sources, 29.98% of conflicts originate from the news caption, 36.86% originate from the news image, and 33.16% from world knowledge. Please refer to Appendix H for the prompts and validation protocols.

### 4.2. Modality Bridging Pre-Training

Although the sources of conflict span both visual and textual modalities, their annotated form in CAC is uniformly text. Therefore, accurately mapping conflict descriptions that originate from vision but exist in text form back to the visual space becomes a challenge. To bridge this modality gap, we introduce a concise and efficient Cross-modal Aligner and forge its cross-modal alignment capability through a dedicated pre-training stage. After the model acquires reliable alignment capabilities, we then commence the second stage of conflict perception training on CAC.

This stage of training is conducted on 50k samples from the FineHARD (Xie et al., 2025) dataset, whose samples

consist of an image $I$, a positive sample $PS$ that exists in the image, and three hard negative samples $\{NS_i\}_{i=1}^3$ that are semantically close to $PS$ but do not exist in the image (see Appendix for examples). For feature extraction, the image $I$ is passed through the vision encoder $\mathcal{E}_V$ and a modality connector $\mathcal{P}$ to obtain the visual feature sequence $V = \mathcal{P}(\mathcal{E}_V(I))$. Simultaneously, the positive and negative text samples, each appended with an <EOS> token, are fed into the LLM. We then extract the hidden state corresponding to the <EOS> token from the LLM's final layer as the global feature. From this, we get the global text feature for the positive sample $\mathbf{t}_p$, and a set of global text features for the negative samples, $\{\mathbf{t}_{n_i}\}$. Next, using $\mathbf{t}_p, \{\mathbf{t}_{n_i}\}$ as the Query and $V$ as the Key and Value (Vaswani et al., 2017), we compute text-guided visual features $\mathbf{v}_p, \{\mathbf{v}_{n_i}\}$ via a Cross-modal Aligner, which is simply implemented as a cross-attention layer:

$$\mathbf{v}_p = \text{Aligner}\left(\mathbf{t}_p, V, V\right), \{\mathbf{v}_{n_i}\} = \text{Aligner}\left(\{\mathbf{t}_{n_i}\}, V, V\right). \tag{1}$$

To achieve a fine-grained alignment, we adopt the contrastive learning loss proposed by SigLIP (Zhai et al., 2023), which aims to ensure that the extracted visual feature $\mathbf{v}_p$ is semantically highly correlated with the corresponding text feature $\mathbf{t}_p$:

$$\mathcal{L}_{cl} = \sum_{(\mathbf{t},\mathbf{v}) \in Q} \frac{1}{1 + e^{y_\mathbf{t}(s_1 \cdot \langle \mathbf{t},\mathbf{v} \rangle + b_1)}}, \tag{2}$$

where $Q = \{(\mathbf{t}_p, \mathbf{v}_p), \{(\mathbf{t}_{n_i}, \mathbf{v}_{n_i})\}, \{(\mathbf{t}_{n_i}, \mathbf{v}_p)\}\}$, $s_1$ and $b_1$ are learnable scalar parameters, and $\langle \cdot, \cdot \rangle$ represents cosine similarity. When $(\mathbf{t},\mathbf{v}) = (\mathbf{t}_p, \mathbf{v}_p)$, $y_\mathbf{t} = 1$; otherwise, $y_\mathbf{t} = -1$.

To aid fine-grained multimodal understanding and preserve the model's inherent language capabilities, we further de-

*Table 2.* Performance comparison (ACC) on multiple datasets using 100-750 (100-350) samples.

| Method | DGM$^4$ | | | | MDSM | | | | MMFakeBench | | | | NewsCLIPpings | | | |
|---|---|---|---|---|---|---|---|---|---|---|---|---|---|---|---|---|
| | 100 | 200 | 500 | 750 | 100 | 200 | 500 | 750 | 100 | 200 | 300 | 350 | 100 | 200 | 500 | 750 |
| Qwen3VL-235B | 51.6 | 51.6 | 51.6 | 51.6 | 56.2 | 56.2 | 56.2 | 56.2 | 57.4 | 57.4 | 57.4 | 57.4 | 61.3 | 61.3 | 61.3 | 61.3 |
| Gemma3-27B | 49.3 | 49.3 | 49.3 | 49.3 | 52.9 | 52.9 | 52.9 | 52.9 | 53.7 | 53.7 | 53.7 | 63.7 | 58.8 | 58.8 | 58.8 | 58.8 |
| LLaMA3.2-90B | 48.3 | 48.3 | 48.3 | 48.3 | 50.8 | 50.8 | 50.8 | 50.8 | 54.1 | 54.1 | 54.1 | 54.1 | 57.9 | 57.9 | 57.9 | 57.9 |
| SeedVL-1.5 | 50.8 | 50.8 | 50.8 | 50.8 | 57.2 | 57.2 | 57.2 | 57.2 | 60.4 | 60.4 | 60.4 | 60.4 | 62.4 | 62.4 | 62.4 | 62.4 |
| HAMMER | 45.3 | 49.8 | 53.8 | 56.9 | 48.1 | 53.9 | 59.8 | 62.2 | 59.7 | 64.7 | 67.1 | 68.2 | 54.4 | 56.2 | 57.4 | 57.3 |
| HAMMER++ | 45.8 | 49.8 | 53.7 | 57.1 | 48.0 | 54.0 | 60.0 | 62.2 | 59.5 | 64.9 | 67.0 | 68.3 | 54.5 | 56.0 | 57.4 | 57.3 |
| RamDG | 46.4 | 50.3 | 55.0 | 57.9 | 48.9 | 53.7 | 58.7 | 61.7 | 60.9 | 64.4 | 68.1 | 68.2 | 55.3 | 56.9 | 57.1 | 57.3 |
| FKA-Owl | 47.4 | 51.0 | 51.2 | 52.4 | 41.3 | 43.8 | 45.3 | 51.1 | 49.9 | 53.2 | 58.4 | 59.1 | 45.4 | 48.0 | 49.8 | 50.0 |
| AMD | 37.9 | 40.5 | 40.7 | 40.8 | 38.2 | 40.7 | 43.9 | 51.3 | 48.3 | 51.4 | 53.9 | 55.2 | 40.4 | 42.1 | 45.6 | 46.7 |
| Qwen2.5VL-3B | 48.0 | 49.7 | 53.4 | 56.3 | 50.3 | 53.2 | 58.1 | 60.3 | 60.2 | 62.4 | 65.3 | 66.5 | 49.8 | 51.9 | 57.2 | 60.4 |
| Gemma3-4B | 48.3 | 50.1 | 54.3 | 60.8 | 49.8 | 52.0 | 57.7 | 63.0 | 61.1 | 64.3 | 66.4 | 66.7 | 48.9 | 51.0 | 57.0 | 61.4 |
| **CORE$_{Qwen}$** | 59.7 | **63.9** | **65.2** | 65.4 | **69.0** | **70.1** | 74.1 | 74.5 | 73.5 | 76.4 | **79.4** | **79.4** | **64.3** | **70.7** | **70.9** | **71.0** |
| **CORE$_{Gemma}$** | **61.3** | 61.9 | 65.2 | **68.4** | 68.4 | 70.0 | **79.3** | **82.0** | **75.2** | **76.7** | 78.1 | 78.1 | 63.0 | 68.4 | 69.6 | 70.9 |
| Δ (vs Best Baseline) | +9.7 | +12.3 | +10.2 | +7.6 | +11.8 | +12.9 | +19.3 | +19.0 | +14.1 | +11.8 | +11.3 | +11.1 | +1.9 | +8.3 | +8.5 | +8.6 |

vise an object-occurrence-based visual question answering task. Specifically, we construct the following question-answering (Antol et al., 2015) instruction format:

**Question:** "Does the image contain RandSF($\{PS, NS_1, NS_2, NS_3\}$)?"

**Answer:** "The image contains $PS$ and doesn't contain $\{NS_1, NS_2, NS_3\}$."

where RandSF($\cdot$) is the random shuffling operation. We then calculate a language generation loss $\mathcal{L}_{o2vqa}$. The total loss function for this stage is defined as follows:

$$\mathcal{L}_{mbpt} = \mathcal{L}_{cl} + \mathcal{L}_{o2vqa}. \qquad (3)$$

### 4.3. Conflict Perception Training

Given a sample $< I, T, \{C_1, C_2\}, \{S_1, S_2\} > \in$ CAC, the news image $I$ is passed through $\mathcal{E}_V$ and $\mathcal{P}$ to obtain the visual feature sequence $V$. The two conflict factors $C_1, C_2$, after appending the <EOS> token, are fed into the LLM to obtain their corresponding global features $\mathbf{t}_{c_1}, \mathbf{t}_{c_2}$. We process the conflict factor features based on the modality of their sources $\{S_1, S_2\}$. If the source $S_i$ of a conflict factor $C_i$ is the visual modality (i.e., $S_i$ is image), we invoke the Cross- modal Aligner pre-trained in Modality Bridging Pre-Training (MBPT) stage to extract the corresponding visual feature. Otherwise, we keep the feature unchanged. For ease of notation, we denote the two conflict representations used for comparison as $\mathbf{z}_1$ and $\mathbf{z}_2$, defined as follows:

$$\mathbf{z}_i = \begin{cases} \text{Aligner}\,(\mathbf{t}_{c_i}, V, V) & \text{if } S_i \text{ is image,} \\ \mathbf{t}_{c_i} & \text{otherwise.} \end{cases} \qquad (4)$$

First, we adopt a conflict-aware contrastive loss $\mathcal{L}_{cacl}$ to help the model establish clear conceptual boundaries by pushing the two conflict factor representations $\mathbf{z}_1$ and $\mathbf{z}_2$ far apart in the semantic space, which is the core to identify conflicts in manipulated samples:

$$\mathcal{L}_{cacl} = \frac{1}{1 + e^{-(s_2 \cdot \langle \mathbf{z}_1, \mathbf{z}_2 \rangle + b_2)}}. \qquad (5)$$

This loss function aims to maximize the distance between the two conflict factor representations.

Besides, we further design a conflict reasoning loss to enhance the conflict capture and preserve the model's inherent language capabilities:

**Question:** "Does the news Real or Fake? If it's fake, further give the reason."

**Answer:** "Real. / Fake. Because the $C_1$ from $S_1$ conflicts with $C_2$ from $S_2$."

We then calculate a language modeling loss $\mathcal{L}_{cr}$ to produce conflict reasoning. The total loss function for CPT stage is defined as follows:

$$\mathcal{L}_{cpt} = \mathcal{L}_{cacl} + \mathcal{L}_{cr}. \qquad (6)$$

### 4.4. Rapid Adaptation to Novel Manipulation

To verify whether the model improved by CORE framework have a clear conceptual understanding, we re-examine the concept pairs of "US President vs. Football Award" and "US President vs. UK Prime Minister". As shown in Figure 2b, the boundaries of conceptual embedding from the model become clear. The MLLMs with rich real-world knowledge and clear conceptual understanding hold good ability for

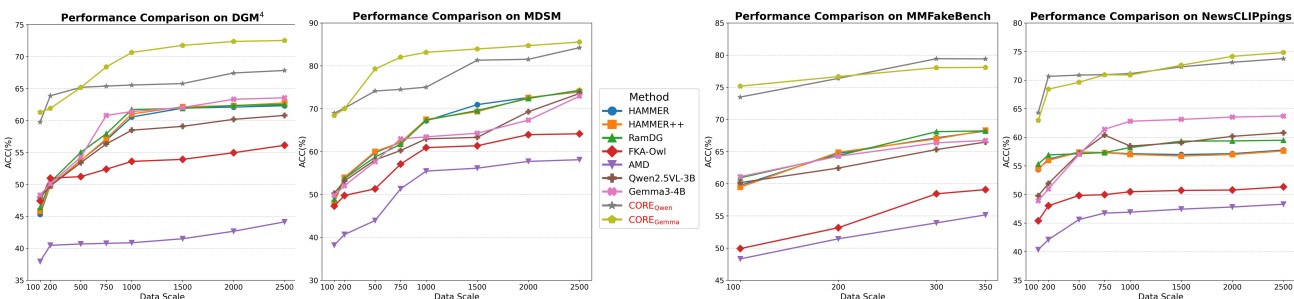

*Figure 5.* Performance comparison on multiple datasets using 100-2.5k (100-350 on MMFakeBench) samples.

*Table 3.* Performance comparison (ACC) on large-scale data.

| Method | SAMM | MDSM | NewsCLIPpings |
|---|---|---|---|
| HAMMER | 92.26 | 86.09 | 64.22 |
| HAMMER++ | 92.43 | 86.31 | 63.95 |
| RamDG | 94.66 | 87.42 | 66.79 |
| FKA-Owl | 92.60 | 87.15 | 68.24 |
| AMD | 87.53 | 80.49 | 60.45 |
| Qwen2.5VL-3B | 92.49 | 87.23 | 71.43 |
| Gemma3-4B | 93.11 | 87.07 | 73.65 |
| **CORE_Qwen** | 96.74 | **88.49** | 81.87 |
| **CORE_Gemma** | **97.14** | 87.61 | **83.45** |

conflict-capturing in manipulated multimodal misinformation. Therefore, when facing with new manipulation types or news patterns, it only requires a small amount of data for fine-tuning to adapt quickly and achieve superior recognition performance. To ensure the model's generalization ability, we do not make specialized designs for specific data types. We only have the model make predictions by constructing a question-answering instruction, *"Is the news real or fake?"*, and calculate the corresponding language generation loss $\mathcal{L}_{ra}$ when finetuning. Similarly, during inference, the model requires only this simple instruction to make predictions.

## 5. Experiments

**Implementation Details.** To validate the generalization ability of the proposed CORE framework, we select two advanced open-source MLLMs as our backbones: Qwen2.5VL-3B (Bai et al., 2025) and Gemma3-4B (Team, 2025b), and apply the CORE framework to train them. All our training processes utilize the LoRA (Hu et al., 2022) technique. Please refer to the Appendix A for more details.

**Datasets.** To comprehensively evaluate the model's performance, we select four public multimodal datasets with diverse manipulation patterns: DGM[4] (Shao et al., 2023), MDSM (Zhang et al., 2025a), MMFakeBench (Liu et al., 2025), and NewsCLIPpings (Luo et al., 2021). To simulate the real-world scenario where novel manipulated data is scarce, we randomly sample a small number of samples

from the aforementioned datasets for training. It is worth noting that, since MMFakeBench (Liu et al., 2025) does not include a training set, we use its validation set as training data. Please refer to the Appendix L for the data overlap analysis between FineHARD, CAC, and the benchmarks.

**Baselines.** For a comprehensive comparison, we select various advanced Multimodal Manipulation Detection methods as baselines and compared their performance against CORE_Qwen and CORE_gemma on multiple datasets. The baseline models include non-MLLMs: HAMMER (Shao et al., 2023), HAMMER++ (Shao et al., 2024), RamDG (Shen et al., 2025); as well as MLLMs specialized for detection tasks: FKA-Owl (Liu et al., 2024), AMD (Zhang et al., 2025a), Qwen2.5VL-3B (Bai et al., 2025), and Gemma3-4B (Team, 2025b). In addition, we also introduce general-purpose MLLMs with larger parameter scales: Qwen3VL-235B (Bai et al., 2023), Gemma3-27B (Team, 2025b), LLaMA-3.2-Vision-90B (Team, 2024), and SeedVL-1.5 (Guo et al., 2025) for zero-shot.

### 5.1. Performance Comparison

**Rapid Adaptation with a Few Samples.** With the conflict-capturing ability of CORE framework, the model can achieve a rapid adaption to novel manipulations with limited data. To verify this, we construct training subsets by randomly sampling 100, 200, 500, 750, 1k, 1.5k, 2k, and 2.5k samples from each dataset, respectively. Table 2 shows the performance of all methods with training set sizes of 100, 200, 500 and 750. Figure 5 shows the performance trend as the training set size increases (100-2.5k).

As shown in Table 2 and Figure 5, both CORE_Qwen and CORE_Gemma achieve SOTA performance across multiple datasets when training data is limited, surpassing the second-best methods by an average of **9.94%** and **10.50%**, respectively. This demonstrates that the CORE framework can rapidly adapt to novel image manipulation techniques. Furthermore, the strong performance of CORE across two different MLLMs validates the method's generalization ability.

**Zero-shot Cross-Manipulation Detection.** To evaluate model's generalization, we utilize the MDSM and DGM[4] datasets, as both feature multiple manipulation categories,

*(a)* Zero-shot performance comparison.

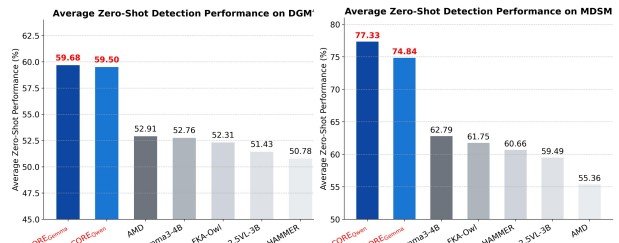

*(b)* Data scale discussion in MBPT stage with $CORE_{Qwen}$.

| Volume | $DGM^4$ | MDSM | MMFakeBench | NewsCLIPpings |
|---|---|---|---|---|
| MBPT-0k | 60.42 | 69.95 | 75.44 | 65.96 |
| MBPT-25k | 63.13 | 71.02 | 77.42 | 68.19 |
| MBPT-**50k** | **65.18** | **74.13** | **79.44** | **70.88** |
| MBPT-75k | 65.12 | 74.03 | 79.48 | 70.92 |

*(c)* Impact of different loss in MBPT with $CORE_{Qwen}$.

| Loss Type | $DGM^4$ | MDSM | MMFakeBench | NewsCLIPpings |
|---|---|---|---|---|
| MSE | 62.52 | 71.03 | 76.42 | 68.97 |
| **Contrastive** | **65.18** | **74.13** | **79.44** | **70.88** |

*(d)* Data scale discussion in CPT stage with $CORE_{Qwen}$.

| Volume | $DGM^4$ | MDSM | MMFakeBench | NewsCLIPpings |
|---|---|---|---|---|
| CPT-0K | 60.51 | 70.29 | 74.62 | 65.91 |
| CPT-7k | 64.74 | 73.20 | 78.96 | 69.57 |
| CPT-**19k** | **65.18** | **74.13** | **79.44** | **70.88** |
| CPT-38k | 64.93 | 73.63 | 78.92 | 70.42 |

*Table 4.* **Ablation and discussion experiments** of (a) average zero-shot performance comparison on $DGM^4$ and MDSM, (b) impact of data scale in MBPT stage, (c) impact of different loss types in MBPT stage, and (d) the impact of data scale in CPT stage.

each with a sufficient number of training samples. In our experimental design, for each specific (target) manipulation type within a dataset, we train a model using 1k randomly sampled instances composed of other manipulation types and authentic news samples. Subsequently, the model is subjected to direct zero-shot inference on a test set constructed exclusively from the held-out target manipulation type and authentic samples. As shown in Table 4a, our methods significantly outperform baselines (e.g., $CORE_{Qwen}$ and $CORE_{Gemma}$ surpass the second-best methods on MDSM by over **14%** and **12%**, respectively). This superiority validates CORE's design principle of detecting the inherent conflicts essential to fake news. Please refer to the Appendix J for Cross-Dataset Zero-Shot.

**Large-scale Training Data.** To verify CORE's scalability, we evaluate it on the full MDSM and SAMM datasets, together with 200k randomly sampled NewsClippings examples.

As shown in Table 3, CORE remains highly effective under large-scale manipulation data of the same type. Across the three benchmarks, $CORE_{Qwen}$ and $CORE_{Gemma}$ outperform the second-best method by an average of **3.79%** and **4.16%**, respectively, demonstrating strong stability and robustness.

**Further Discussion.** Appendix I reports experiments on time-sensitive events beyond the model's pre-training scope, while Appendix K compares prompting strategies with CORE training.

## 5.2. Ablation Study

To validate the effectiveness of each component within the CORE framework, We designed a series of ablation experiments on $CORE_{Qwen}$:

**Data Scale in MBPT.** We evaluated the impact of MBPT data volume (0–75k) on random subsets (500 (300) samples). As shown in Table 4b, insufficient data (0k, 25k) leads to an under-trained Cross-modal Aligner, hindering conflict localization and causing a **4.47%** average drop (0k vs. 50k). Conversely, the negligible gap between 50k and 75k indicates performance saturation, confirming that 50k samples suffice for robust alignment.

**Loss Discussion in MBPT.** To validate our loss design, we compared it with an MSE-based alternative where ROIAlign (He et al., 2017) aligns positive visual features $PS$ directly with text features $v_{c_i}$. As shown in Table 4c, this explicit supervision lags behind contrastive learning (He et al., 2020) by **2.67%**. This decline stems from disrupting the training consistency with CPT and ROIAlign's limitation in capturing global semantics. For instance, resolving conflicts (e.g., linking "Ballon d'Or" to a "football field") requires global context that ROIAlign's strict localization severs, thereby degrading performance.

**Data Scale in CPT.** To explore the impact of data scale in CPT, we adjusted the CAC volume from 0k to 38k compared to the standard setting (19k). The results in Table 4d reveal that removing the CPT stage (0k) leads to a sharp performance deterioration (avg. **4.58%**), underscoring the necessity of CPT. Increasing the volume to 7k improves accuracy but still lags behind 19k (avg. **0.79%** gap), suggesting insufficient data for learning semantic conflicts. Conversely, expanding to 38k slightly degrades performance (avg. **0.43%**), potentially due to overfitting on manipulation artifacts specific to the SAMM dataset.

**Loss Components in MBPT and CPT.** To further disentangle the contribution of each objective, we remove one loss term in MBPT and CPT, as shown in Table 5. In MBPT,

*Table 5.* **Loss component ablation** in MBPT and CPT stages on $\text{CORE}_{\text{Qwen}}$.

*(a)* Ablation on MBPT loss.

| Loss Combination | DGM$^4$ | MDSM | MMFakeBench | NewsCLIPpings |
|---|---|---|---|---|
| w/o $\mathcal{L}_{cl}$ | 61.01 | 69.33 | 75.52 | 66.03 |
| w/o $\mathcal{L}_{o2vqa}$ | 63.93 | 73.12 | 77.99 | 69.12 |
| **Full** ($\mathcal{L}_{cl} + \mathcal{L}_{o2vqa}$) | **65.18** | **74.13** | **79.44** | **70.88** |

*(b)* Ablation on CPT loss.

| Loss Combination | DGM$^4$ | MDSM | MMFakeBench | NewsCLIPpings |
|---|---|---|---|---|
| w/o $\mathcal{L}_{cacl}$ | 55.17 | 59.30 | 68.94 | 59.22 |
| w/o $\mathcal{L}_{cr}$ | 63.77 | 73.22 | 78.10 | 69.91 |
| **Full** ($\mathcal{L}_{cacl} + \mathcal{L}_{cr}$) | **65.18** | **74.13** | **79.44** | **70.88** |

removing $\mathcal{L}_{cl}$ causes a larger average drop than removing $\mathcal{L}_{o2vqa}$ (**4.44%** vs. **1.37%**), indicating that contrastive alignment provides the primary signal for bridging textual concepts and visual evidence, while the VQA-style objective further stabilizes multimodal instruction learning. In CPT, removing $\mathcal{L}_{cacl}$ leads to the most severe degradation (avg. **11.75%**), confirming that explicitly separating conflicting factors is central to conflict perception. Removing $\mathcal{L}_{cr}$ also reduces performance (avg. **1.16%**), showing that conflict reasoning supervision helps preserve the model's ability to express and use the learned conflict cues during prediction.

## 6. Conclusion

This paper introduces CORE, a conflict-oriented reasoning framework that enhances MLLMs with explicit conflict-capturing capability for robust and generalizable misinformation detection. By leveraging the newly constructed CAC and a conflict-aware training paradigm, CORE effectively conducts a conflict-perception training and enables rapid adaptation to unseen manipulation types.

## Impact Statement

**Positive Impacts:** By focusing on fundamental inconsistencies rather than specific manipulation patterns, CORE improves the robustness of information ecosystems against evolving generative AI. Its high-efficiency reasoning reduces the energy-intensive requirement for frequent full-parameter retraining, supporting sustainable AI deployment.

**Ethical Considerations & Mitigation:** To prevent the potential misuse of our model or the Conflict Attribution Corpus (CAC) for generating more deceptive content, we will implement a strict data access protocol. The CAC will be released exclusively for pure research purposes under a specialized license that prohibits its use in generative tasks. Furthermore, all data in the corpus are sourced from public domains to safeguard privacy.

## Acknowledgements

This work was funded by the National Natural Science Foundation of China (No. 62572166, 62302140, 62502144, 62502142, 62573399) and the Natural Science Foundation of Anhui Province (No. 2508085QF226). The computation is completed on the HPC Platform of Hefei University of Technology.

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

## A. Implementation Details

During MBPT, we optimize the Modality Connector and the LLM with a uniform learning rate of $1 \times 10^{-4}$ for 1 epoch, while the Vision Encoder is frozen. In CPT, we continue to optimize the Modality Connector and the LLM for 3 epochs, while the Vision Encoder remains frozen. In RA, we optimize only the LLM for 3 to 8 epochs, depending on the scale of the training data. All experiments are conducted on devices equipped with 4 NVIDIA H200 GPUs.

## B. Benchmark Brief Introduction

To ensure a rigorous evaluation of generalization, we select multiple representative benchmarks that cover a wide spectrum of manipulation types. The defining features of these datasets are outlined below:

**NewsCLIPpings** focuses on the out-of-context (OOC) threat scenario: it provides a dataset where both the image and text are individually unmanipulated, but are automatically mismatched to create semantic or entity inconsistencies.

**DGM**[4] employs a random manipulation pipeline: it randomly manipulates a part of the news's image or caption (e.g., randomly replacing some words in the caption with other words).

**MMFakeBench** introduces a comprehensive benchmark for mixed-source MMD: it includes 3 critical sources (textual veracity distortion, visual veracity distortion, and cross-modal consistency distortion) along with 12 sub-categories of misinformation forgery types.

**SAMM** pioneers the detection of semantically-coordinated manipulations: it first applies SOTA image manipulations, and then generates contextually-plausible, semantically consistent textual narratives designed to reinforce the visual deception.

**MDSM** utilizes an adversarial pipeline that leverages MLLMs to simulate high-risk disinformation: it first alters images using SOTA editing techniques, and then pairs them with MLLM-generated deceptive texts that maintain semantic consistency with the visual manipulations.

## C. Concepts in Section 3

---

**UK Prime Minister**

**#Textual Concepts**

["Chancellor of the Exchequer", "Home Secretary", "Foreign Secretary", "Despatch Box", "Lord Chancellor", "Secretary of State for Health and Social Care", "Secretary of State for Education", "Leader of the House of Commons", "Leader of the House of Lords", "Chief Whip", "Cabinet Office", "10 Downing Street", "Chequers", "Prime Minister's Questions (PMQs)", "Cabinet Reshuffle", "Cabinet Secretary", "Downing Street Press Secretary", "The King's Speech", "Special Adviser (SpAd)", "Prime Minister's Private Office", ... ]

**#Visual Concepts Images**

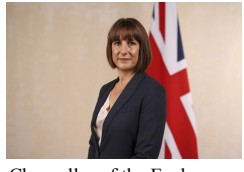 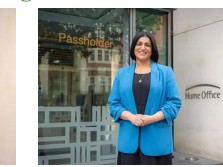 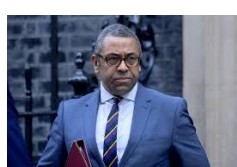 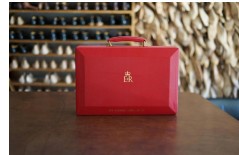 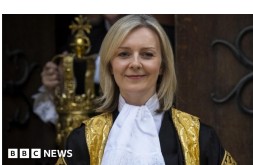

Chancellor of the Exchequer    Home Secretary    Foreign Secretary    Despatch Box    Lord Chancellor

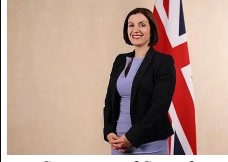 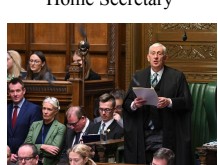 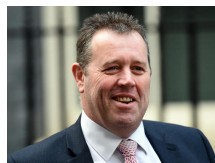 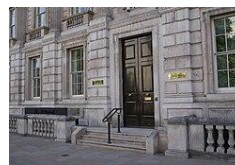 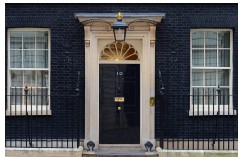

Secretary of State for Education    Leader of the House of Commons    Chief Whip    Cabinet Office    10 Downing Street

---

*Figure 6.* Examples of UK Prime Ministers.

## US President

**#Textual Concepts**

["Camp David", "Hail to the Chief", "Seal of the President of the United States", "Secretary of State", "Secretary of Defense", "Designated Survivor", "White House Chief of Staff", "Twenty-fifth Amendment", "Nuclear Football", "Bully Pulpit", "Executive Order", "White House Chief Usher", "Presidential Daily Brief", "Presidential Library", "Electoral College", "Inauguration Day", "Presidential Oath of Office", "West Wing", "East Wing", "Situation Room", "Resolute Desk", ...]

**#Visual Concepts Images**

| | | | | |
|---|---|---|---|---|
| Seal of the President of the United States | Camp David | Hail to the Chief | Secretary of State | Secretary of Defense |
| Designated Survivor | White House Chief of Staff | Nuclear Football | Executive Order | Presidential Daily Brief |

*Figure 7.* Examples of US President.

## Football Award

**#Textual Concepts**

["The Best FIFA Men's Player", "The Best FIFA Women's Player", "Ballon d'Or Féminin", "UEFA Men's Player of the Year Award", "UEFA Women's Player of the Year Award", "Copa América", "UEFA European Championship", "Africa Cup of Nations", "AFC Asian Cup", "CONCACAF Gold Cup", "Copa Libertadores", "UEFA Europa League", "CAF Champions League", "AFC Champions League", "CONCACAF Champions Cup", "FIFA Club World Cup", "Premier League", "La Liga", "Serie A", "Bundesliga", ...]

**#Visual Concepts Images**

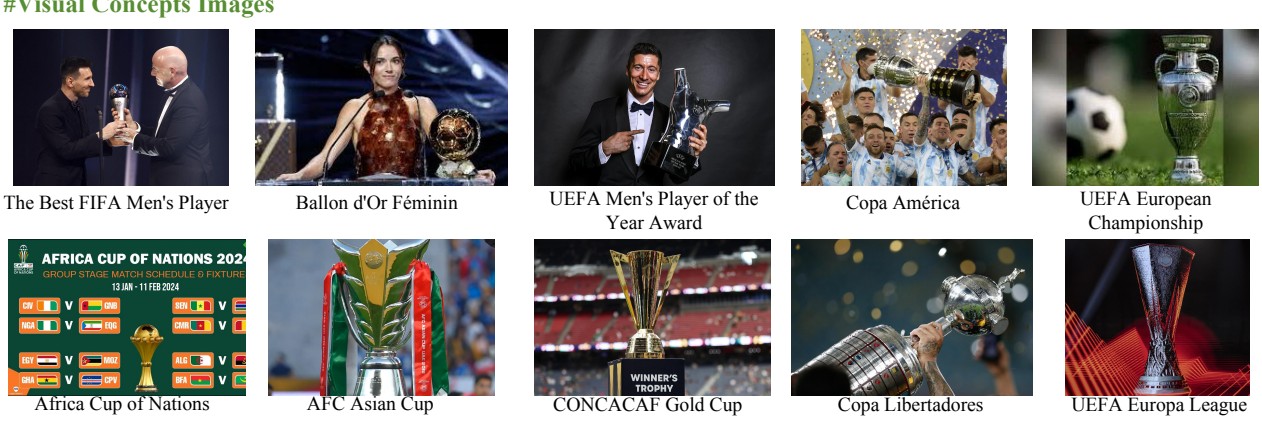

| | | | | |
|---|---|---|---|---|
| The Best FIFA Men's Player | Ballon d'Or Féminin | UEFA Men's Player of the Year Award | Copa América | UEFA European Championship |
| Africa Cup of Nations | AFC Asian Cup | CONCACAF Gold Cup | Copa Libertadores | UEFA Europa League |

*Figure 8.* Examples of Football Award.

Figure 6-8 illustrates some of the textual concepts and images used in our experiments in Section 3.

## D. Examples of CAC

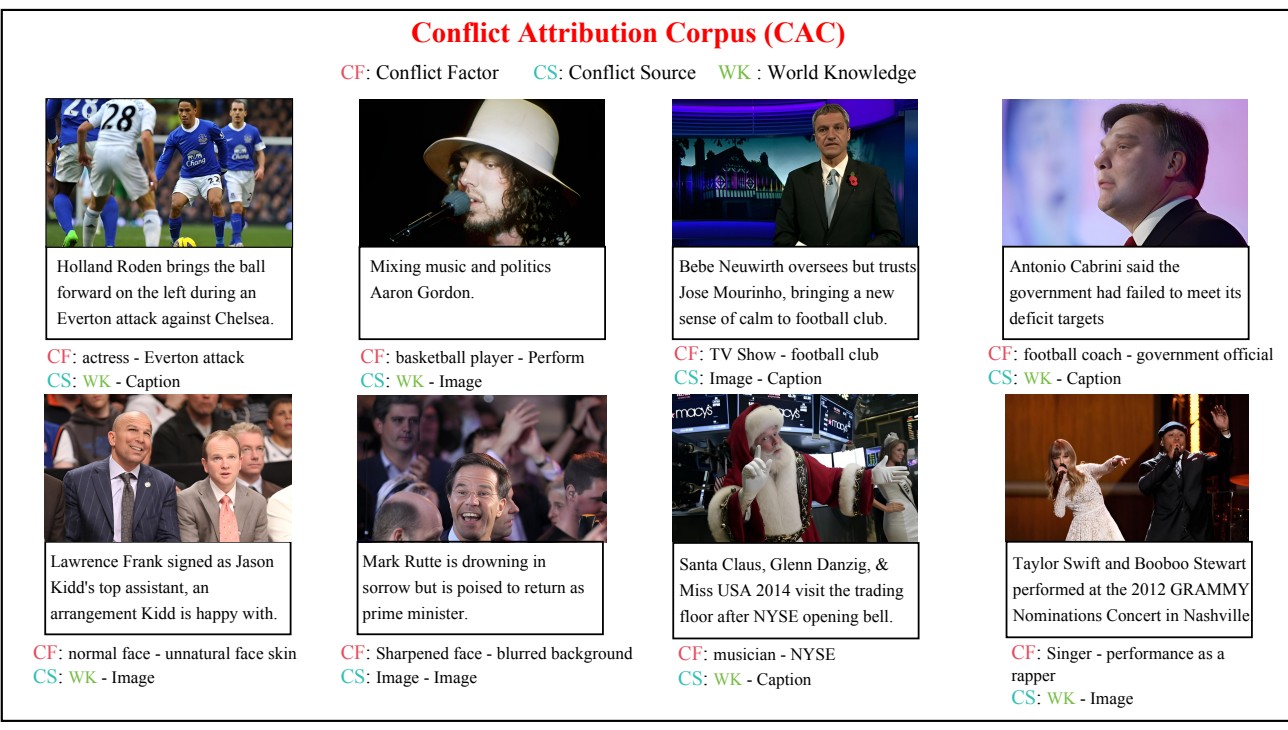

*Figure 9.* Examples of CAC

## E. Examples of FineHARD

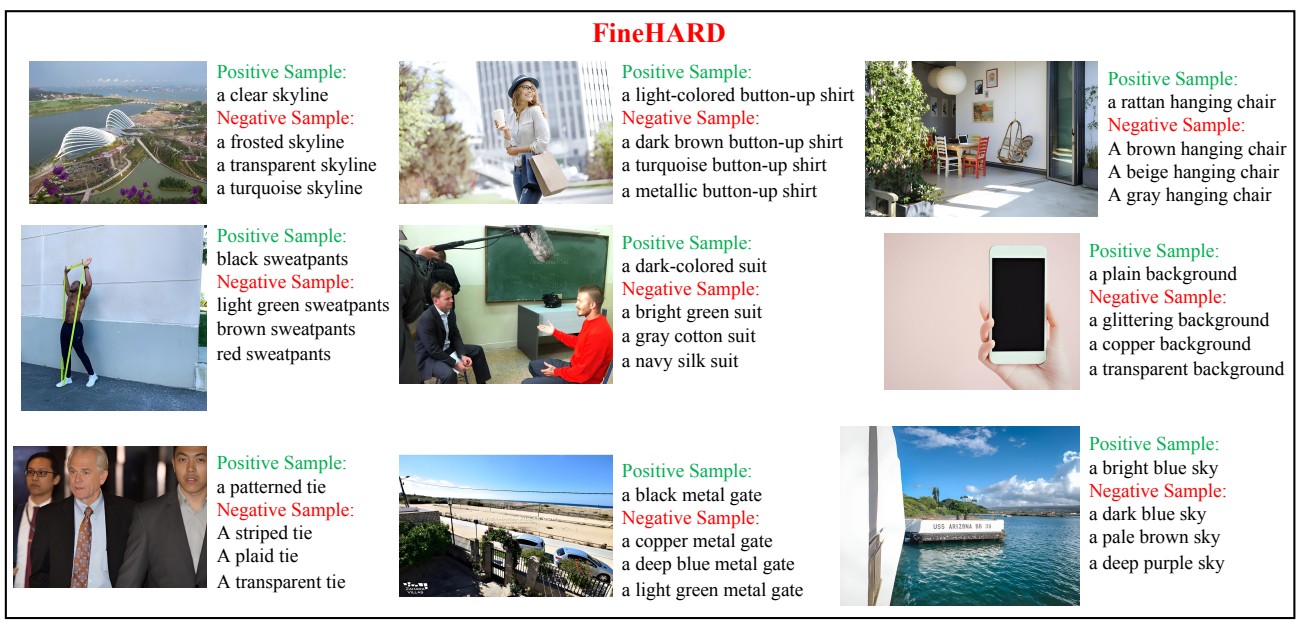

*Figure 10.* Examples of FineHARD.

# F. Feature Extraction Details in Section 3

For visual features, the image is passed through the vision encoder $\mathcal{E}_V$ and the modality connector $\mathcal{P}$ to obtain a sequence of visual feature embeddings. For textual features, the text concept is fed into the LLM, and we extract the corresponding sequence of hidden states from its final layer. Subsequently, we apply an averaging operation to both the visual and textual feature sequences, transforming each into a tensor of shape $[1, \text{hidden size}]$.

# G. Construction of World Knowledge Evaluation Benchmark

To assess whether models possess the necessary background knowledge for fake news detection, we constructed the World Knowledge Evaluation Benchmark. This benchmark comprises 200 multiple-choice questions meticulously designed to cover a diverse range of domains frequently targeted by misinformation, including current events (e.g., political leaders, cultural awards), social movements, geography, history, and science.

A key aspect of our construction process was the creation of plausible, semantically-related distractors. Unlike standard QA datasets where incorrect answers are often random, our distractors share the same semantic category as the ground truth. For example, distractors for the question "Which film won the Academy Award for Best Picture in 2024?" include other highly-nominated films from the same ceremony (e.g., "Barbie," "Poor Things"). Similarly, distractors for the 2024 German Chancellor include the former chancellor and other contemporary European leaders. This design ensures the benchmark evaluates precise knowledge rather than coarse categorical association. Figure 11 illustrates several examples from this benchmark.

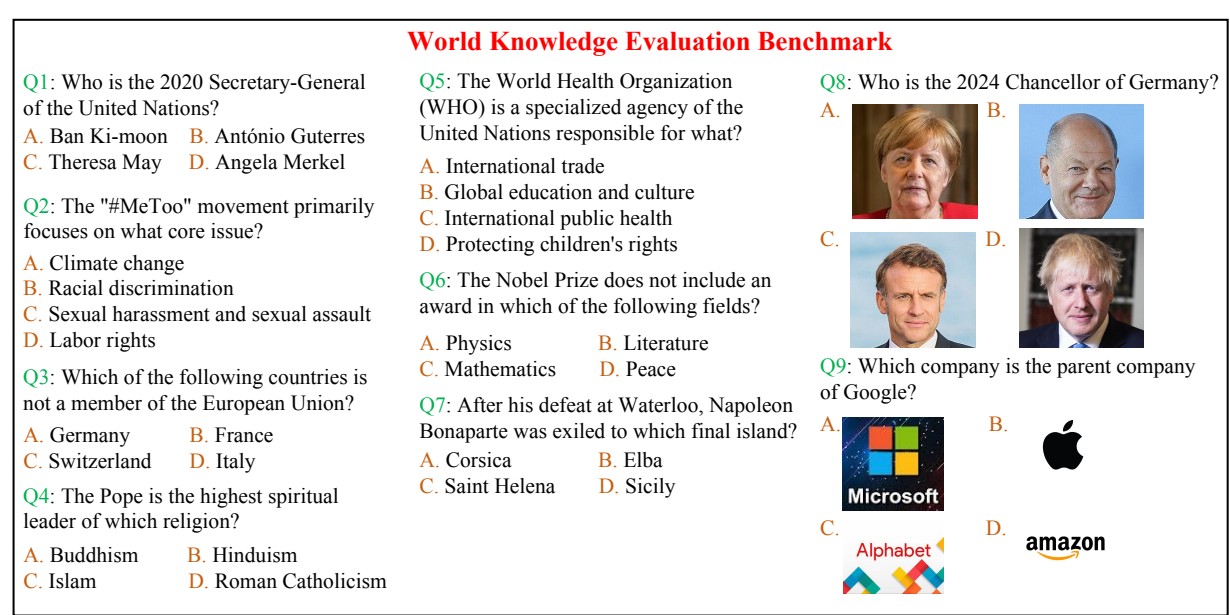

*Figure 11.* Examples of the World Knowledge Evaluation Benchmark, highlighting the use of semantically plausible distractors.

To address the need for transparency regarding data provenance and curation, we provide the specific construction details below, focusing on data sources, collection pipelines, and domain-balancing criteria.

### G.1. Data Sources

To ensure factual accuracy and relevance, we curated data from two primary streams:

- **Authoritative Knowledge Bases:** Static facts (e.g., geography, history, scientific definitions) were sourced from

high-reliability encyclopedic sources, primarily Wikipedia and Britannica, ensuring that the ground truth is indisputable.

- **Verified News Outlets:** Dynamic facts involving current events (e.g., 2024 political leaders, recent cultural awards) were cross-referenced against major reputable news agencies (e.g., BBC, Reuters, AP).

## G.2. Collection and Generation Pipeline

The question collection process followed a "Human-Guided, AI-Assisted" paradigm involving three stages:

1. **Entity and Topic Extraction:** We first identified high-frequency entities and topics appearing in the news, such as political figures (e.g., US Presidents, European leaders), celebrities, and major global organizations (e.g., WHO, UN). This ensures the benchmark evaluates knowledge directly relevant to the downstream detection task.

2. **Distractor-Aware Question Generation:** As noted in the introduction, we enforced the creation of *semantically plausible distractors* to test specific factual knowledge rather than simple elimination capabilities.

3. **Adversarial Filtering:** We employed GPT-4o to review the questions. Any question that could be answered solely through linguistic bias or simple elimination without specific knowledge was discarded or rewritten.

## G.3. Selection and Domain-Balancing Criteria

To prevent domain bias, we established a strict taxonomy covering five distinct categories. The 200 questions were balanced to ensure broad coverage of the "world knowledge" typically exploited in multimodal misinformation. The domain distribution and selection criteria are defined in Table 6.

*Table 6.* Domain Distribution and Selection Criteria for the World Knowledge Benchmark.

| Domain | Ratio | Selection Criteria |
|---|---|---|
| **Current Events** | 30% | Focuses on heads of state (e.g., German Chancellor, UK PM), major elections, and shifting geopolitical alliances active between 2020–2024. |
| **Culture & Entertainment** | 20% | Includes major awards (Oscars, Grammys, Ballon d'Or) and celebrity identities often targeted by "face swap" deepfakes. |
| **Social & Political Movements** | 15% | Covers defining movements (e.g., #MeToo, BLM, Climate Change initiatives) to test understanding of complex social contexts. |
| **History & Geography** | 20% | Tests static knowledge often used as background context in news (e.g., EU membership status, locations of landmarks, historical exiles). |
| **Science & Public Health** | 15% | Focuses on verifiable scientific consensus and major organizations (e.g., WHO mandates, COVID-19 terminology) frequently subject to medical misinformation. |

**Final Human Verification** All questions underwent a final manual verification round by the authors to confirm that: (1) the ground truth is unambiguous, and (2) the distractors are factually incorrect but contextually relevant.

## H. Prompt Templates and Validation Protocols for CAC Construction

To ensure reproducibility and transparency regarding the construction of the Conflict Attribution Corpus (CAC), we provide the exact prompts used at each stage of the pipeline, along with the specific validation criteria employed by the MLLM expert pool and human annotators. As detailed in Section 4.1, our pipeline utilizes an expert pool $\mathcal{M} = \{\text{GPT-4o, Gemini-2.5-Pro, Qwen3-VL-Plus}\}$.

## H.1. Background Knowledge Collection Prompts

As described in the implementation details, this stage leverages an MLLM to bridge the gap between the raw news sample and external knowledge. The process follows a specific pipeline: analyzing the image and caption to extract key semantic information (e.g., time, event, celebrities), combining this into search queries, conducting separate web and image searches via the Google Search API, and finally validating the relevance of retrieved materials.

*Table 7.* Prompts for Background Knowledge Collection (Google Search API Stage).

---

**Step 1: Semantic Extraction and Query Generation**

---

**Input:** Image $I$, Caption $T$, Prior Manipulation Info $P$ (from SAMM).
**Prompt:** "You are an investigative expert assisting in fact-checking. 1. Analyze the provided news image and caption to extract key semantic information, specifically identifying: **Time** (dates, eras), **Events** (political rallies, ceremonies, conflicts), and **Celebrities/Entities** (politicians, organizations). 2. Based on these extracted semantics, generate 3 distinct search queries for the Google Search API to gather external evidence. 3. Ensure queries are suitable for both *Textual Search* (to find news articles) and *Image Search* (to find original source photos). Return the output as a JSON list of query strings."

---

**Step 2: Retrieved Material Relevance Validation**

---

**Input:** Image $I$, Caption $T$, Retrieved Material $R$ (Text Snippet or Image).
**Prompt:** "You are a data filter. Review the retrieved material $R$ obtained from Google Search. Does $R$ provide highly relevant factual context, background details, or visual evidence related to the entities and events extracted from the query image $I$ and caption $T$? **Criterion:** Answer 'YES' only if $R$ is directly relevant and useful for verifying the authenticity of the news; otherwise, answer 'NO'."

---

## H.2. Conflict Rationale Generation Prompts

The core logic of CORE relies on identifying *why* a sample is fake. We task a randomly selected MLLM from $\mathcal{M}$ to generate this rationale, which is then cross-validated by the other experts in the pool.

*Table 8.* Prompts for Conflict Rationale Generation and Validation.

---

**Generation Prompt**

---

**System:** You are a forensics expert specializing in multimodal misinformation.
**Input:** Image $I$, Caption $T$, Validated Background Knowledge $B$ (from Step 2), Prior Manipulation Info $P$.
**Instruction:** "This news item is known to be manipulated based on Prior Info $P$. Using the external Background Knowledge $B$ gathered via search, explain specifically *why* it is false. Focus on the intrinsic conflict between the visual content, the textual caption, and world knowledge. Do not simply state it is fake; describe the logical contradiction in detail."

---

**Validation Prompt**

---

**System:** You are a quality assurance auditor for misinformation detection.
**Input:** Image $I$, Caption $T$, Validated Background Knowledge $B$, Generated Rationale $R$.
**Instruction:** "Evaluate the validity of the provided Rationale $R$ against the evidence. Check two criteria:
1. **Consistency:** Does $R$ accurately reflect the visible content in $I$ and $T$?
2. **Factual Support:** Is the logical contradiction described in $R$ fully supported by the Background Knowledge $B$?
Return 'Valid' only if both criteria are strictly met; otherwise return 'Invalid'."

---

## H.3. Conflict Structuring Prompts

This stage distills the natural language rationale into the structured format $< C_1, C_2, S_1, S_2 >$ required for the CPT stage.

*Table 9.* Prompts for Conflict Structuring and Validation.

**Structuring Prompt**

**Input:** Image $I$, Caption $T$, Approved Rationale $R_{expl}$.
**Instruction:** "Based on the provided rationale, extract the two specific conflicting elements (Conflict Factors) and their origins (Conflict Sources).
Definitions:
- *Conflict Factor (C)*: The specific concept (e.g., 'USA President', 'Ballon d'Or').
- *Conflict Source (S)*: Where $C$ comes from. Must be one of ['Image', 'Caption', 'World Knowledge'].
Output strictly in this JSON format:
{"C1": "...", "S1": "...", "C2": "...", "S2": "..."}"

**Validation Prompt**

**System:** You are a data quality auditor ensuring alignment between textual reasoning and structured data.
**Input:** Approved Rationale $R_{expl}$, Generated JSON Structure $J$.
**Instruction:** "Verify if the JSON object $J$ accurately distills the conflict described in Rationale $R_{expl}$. Check the following:
1. **Content Match:** Do $C_1$ and $C_2$ represent the exact contradictory concepts mentioned in $R_{expl}$?
2. **Source Attribution:** Are the sources $S_1$ and $S_2$ correctly identified as 'Image', 'Caption', or 'World Knowledge' according to the rationale?
Return 'Valid' only if the extraction is precise and the format is correct; otherwise return 'Invalid'."

## H.4. Final Human Verification Protocol

While the automated pipeline ensures scalability, we introduced a rigorous human-in-the-loop verification step to ensure the quality of the CAC dataset. We employed 5 distinct annotators to validate the dataset.

**Sampling Strategy:** We randomly select 1k samples, ensuring balanced coverage across the various conflict source distributions.

**Validation Rubric:** Annotators were instructed to reject or correct samples based on the following criteria:

1. **Conflict Existence:** Does a logical contradiction actually exist between $C_1$ and $C_2$?

2. **Source Accuracy:** Are $S_1$ and $S_2$ correctly attributed? (e.g., if $S_1$ is labeled 'World Knowledge', does it rely on external facts rather than visual cues?)

3. **Granularity Check:** Are the conflict factors fine-grained concepts (e.g., "red tie" vs "blue tie") rather than abstract descriptions (e.g., "fake image" vs "real text")?

**Outcome:** Ultimately, 993 samples passed human verification (a pass rate of 99.3%), indicating high reliability in the automated generation process.

## I. Evaluation on Time-Sensitive Events

To address the concern regarding the model's applicability when handling *time-sensitive events that fall outside the model's pre-trained knowledge scope*, we conducted an additional evaluation focusing on emerging misinformation.

**Dataset Construction.** We collected a distinct dataset consisting of 100 high-risk, time-sensitive fake news samples from social media platforms. To rigorously test the "out-of-scope" condition, the majority of these events occurred in 2025, ensuring they post-date the training cut-off of the foundation models and our training corpus. These samples simulate real-world "zero-day" misinformation scenarios where specific world knowledge is absent from the model's parametric memory. Figure 12 visualizes three examples from this collected dataset.

**Baselines.** We benchmarked CORE$_{\text{Qwen}}$ against three representative state-of-the-art methods: HAMMER, AMD and FKA-Owl.

**Results and Analysis.** The quantitative results are presented in Table 10. While baseline methods struggle significantly due to their reliance on specific patterns or outdated knowledge bases (ranging from 44% to 52% accuracy), CORE achieves an accuracy of 74%.

This superior performance indicates that while CORE$_{\text{Qwen}}$ may lack specific knowledge of the exact 2025 event (e.g., the specific outcome of a new election), its *Conflict-Oriented Reasoning* paradigm allows it to identify falsified news by detecting: (1) **Intrinsic Logic Violations:** Contradictions within the text or between visual elements that violate general physical or logical rules (which remain constant regardless of the year). (2) **Cross-Modal Inconsistencies:** Discrepancies between the provided image and the textual claim that do not require specific entity knowledge to detect (e.g., emotional mismatch, scene inconsistency).

*Table 10.* Performance comparison on Time-Sensitive Fake News. This dataset comprises events falling outside the pre-trained knowledge scope of the models.

| Method | Accuracy (%) |
|---|---|
| AMD | 44.0 |
| HAMMER | 48.0 |
| FKA-Owl | 52.0 |
| **CORE$_{\text{Qwen}}$** | **74.0** |

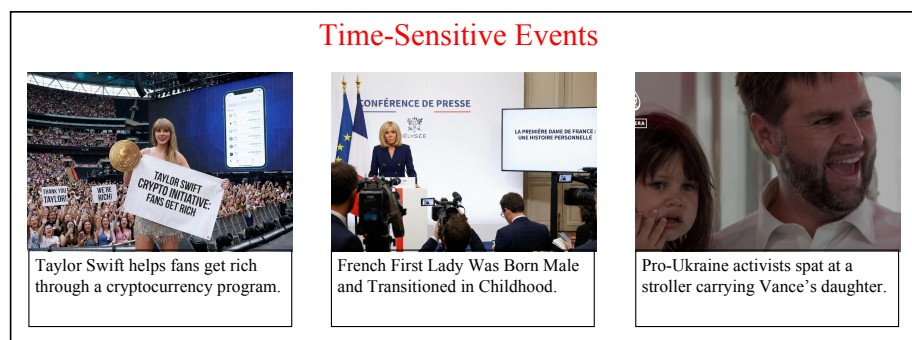

*Figure 12.* Examples of time-sensitive fake news samples (2025) used in our evaluation. Despite lacking specific pre-trained knowledge of these recent events, CORE successfully identifies the misinformation by detecting intrinsic logical conflicts and cross-modal inconsistencies, whereas baseline models often fail.

## J. Cross-Dataset Zero-Shot Generalization

We conduct a rigorous **Cross-Dataset Generalization** experiment. This setting is designed to evaluate the model's performance in a true open-world scenario, where the testing data originates from a completely different source distribution with potentially unseen manipulation pipelines compared to the training data.

**Experimental Setup.** We use four benchmarks: NewsCLIPpings, DGM$^4$, MMFakeBench, and MDSM. Specifically, to evaluate performance on a target dataset (e.g., NewsCLIPpings), we trained the models on a subset composed of the remaining three datasets (e.g., DGM$^4$, MMFakeBench, and MDSM). To simulate a low-resource adaptation scenario effectively, we randomly sampled a total of only 3,000 samples from the union of the three training datasets. The models were then evaluated directly on the full test set of the held-out target dataset in a zero-shot manner.

**Results and Analysis.** The quantitative results are reported in Table 11. As observed, existing methods struggle significantly in this cross-dataset setting. For instance, AMD, which relies heavily on specific manipulation traces and prior information, drops to near-random performance (40.2% - 46.1%) when the testing distribution shifts. Similarly, FKA-Owl and HAMMER exhibit limited robustness, with accuracies hovering between 43% and 54%. This suggests that these baselines tend to overfit to the specific data distributions or manipulation artifacts present in their training domains, failing to generalize to the semantic or physical inconsistencies inherent in unseen datasets.

In contrast, CORE$_{\text{Qwen}}$ demonstrates superior generalization capabilities, consistently outperforming all baselines across all four held-out datasets. Specifically, CORE$_{\text{Qwen}}$ achieves an accuracy of 60.3% on NewsCLIPpings and 63.4% on MDSM,

surpassing the best-performing baseline by margins of **10.0%** and **16.9%**, respectively. On average, our method improves over the second-best approach by approximately **11.4%**. This significant improvement validates that the CORE framework, by focusing on the fundamental "conflict" logic rather than dataset-specific artifacts, successfully equips MLLMs with a more abstract and robust reasoning capability suitable for open-world multimodal misinformation detection.

*Table 11.* Cross-Dataset Zero-Shot Performance (ACC %). Models are trained on a mixed subset (3k samples) of three datasets and evaluated zero-shot on the held-out fourth dataset.

| Method | NewsCLIPpings | DGM$^4$ | MMFakeBench | MDSM |
|---|---|---|---|---|
| HAMMER | 50.3 | 47.2 | 54.1 | 46.5 |
| FKA-Owl | 43.8 | 46.7 | 53.3 | 42.9 |
| AMD | 40.2 | 42.1 | 46.1 | 40.3 |
| **CORE$_{Qwen}$** | **60.3** | **57.3** | **62.6** | **63.4** |

## K. Prompting vs. Training

A natural question arises regarding the contribution of the proposed training framework: *Can MLLMs achieve similar conflict detection capabilities by simply using a similar prompting strategy without the specialized training?*

To answer this, we conducted an experiment where we directly prompted the MLLMs. Specifically, we instructed the models to first generate the conflict factors ($C_1$ and $C_2$) based on the input image and text, and subsequently use these factors to deduce whether the news is real or fake. We applied this prompting strategy to the backbone model Qwen2.5-VL-3B, as well as to two significantly larger and more powerful state-of-the-art MLLMs: Llama-3.2-Vision-90B and Seed-1.6.

We compared these "Prompt-only" baselines against our CORE$_{Qwen}$ fine-tuned on 100 samples (Rapid Adaptation). The results are reported in Table 12.

*Table 12.* Performance comparison between direct prompting and CORE training (100 samples). **Prompt-only** denotes using the model directly with conflict-oriented instructions without MBPT/CPT training. CORE$_{Qwen}$ trains on 100 samples.

| Method | DGM$^4$ | MDSM | MMFakeBench | NewsCLIPpings |
|---|---|---|---|---|
| Qwen2.5-VL-3B | 45.3 | 46.2 | 55.3 | 47.1 |
| Llama-3.2-Vision-90B | 49.7 | 53.5 | 53.8 | 58.6 |
| Seed-VL-1.6 | 52.1 | 59.3 | 62.1 | 64.7 |
| **CORE$_{Qwen}$** | **59.7** | **69.0** | **73.5** | **64.3** |

**Analysis.** As shown in Table 12, directly prompting the models yields suboptimal results compared to the CORE framework, even when using significantly larger models. The prompt-only Qwen2.5-VL-3B achieves only 45.3% on DGM$^4$ and 46.2% on MDSM. In contrast, by training on merely 100 samples, CORE$_{Qwen}$ boosts performance to 59.7% (+14.4%) and 69.0% (+22.8%) respectively. This indicates that without the explicit MBPT and CPT, the model struggles to accurately ground visual concepts and identify subtle contradictions, often hallucinating conflicts or failing to align the visual and textual modalities effectively.

In conclusion, simply instructing an MLLM to "find conflicts" is insufficient. The CORE framework is essential to endow the model with the actual capability to perceive and reason about these conflicts, achieving superior generalization with minimal data.

## L. Data Leakage and Overlap Analysis

To ensure that the reported performance gains reflect genuine generalization rather than dataset bias or memorization, we explicitly clarify the relationship between our training sources (FineHARD and SAMM) and the evaluation benchmarks (NewsCLIPpings, DGM$^4$, MMFakeBench, and MDSM). We conduct both a source-based qualitative analysis and a rigorous CLIP-based empirical verification to rule out data leakage.

## L.1. Source and Distribution Analysis

**1. Pre-training Data (FineHARD vs. Benchmarks):** The FineHARD dataset, used for our pre-training, is constructed based on LAION-2B. This dataset predominantly consists of general-domain natural images and web-crawled captions. In contrast, the evaluation benchmarks (e.g., NewsCLIPpings and DGM[4]) are derived primarily from the VisualNews dataset, which focuses strictly on news events and journalistic imagery. There is a fundamental domain gap between the general "in-the-wild" distribution of LAION-2B and the specific "news-caption" distribution of the benchmarks, minimizing the likelihood of direct overlap.

**2. CAC (SAMM vs. Benchmarks):** The SAMM dataset, utilized for the Conflict-Aware Contrastive (CAC) learning, employs a distinct generation pipeline for creating manipulations. The benchmarks (e.g., DGM[4] and NewsCLIPpings) rely on manipulation techniques that differ significantly from the patterns in SAMM (See Appendix B). Consequently, there is no overlap in the manipulation logic or the specific samples used.

## L.2. Empirical Verification via CLIP Similarity

To further quantitatively verify the absence of overlap, we conducted a comprehensive similarity search across the datasets.

**Methodology.** We employed a pre-trained CLIP model to extract features from:

- The specific subsets of the FineHARD (Used in MBPT) and SAMM (CAC) datasets that are actually utilized for training.

- The full test sets of all four benchmarks: NewsCLIPpings, DGM[4], MMFakeBench, and MDSM.

For every pair of samples $(S_{\text{train}}, S_{\text{test}})$—where $S_{\text{train}}$ is from the training sources and $S_{\text{test}}$ is from the benchmarks—we calculated a composite similarity score $\text{Score}_{\text{sim}}$. This score is defined as the sum of four cross-modal and uni-modal cosine similarities:

$$\text{Score}_{\text{sim}} = \text{Sim}(I_{\text{train}}, I_{\text{test}}) + \text{Sim}(T_{\text{train}}, T_{\text{test}}) + \text{Sim}(I_{\text{train}}, T_{\text{test}}) + \text{Sim}(T_{\text{train}}, I_{\text{test}}) \tag{7}$$

where $I$ and $T$ represent the image and text embeddings, respectively.

**Results.** We identified and retrieved the top-200 pairs with the highest $\text{Score}_{\text{sim}}$ from the millions of potential combinations. A manual inspection of these top-200 pairs was conducted. The inspection revealed that even among the pairs with the highest similarity scores, there were no identical images or captions, and no content duplication was observed. The matches were primarily based on broad semantic similarities (e.g., two different images containing a "dog" or a "politician") rather than data leakage.

**Conclusion.** Both the source provenance analysis and the empirical feature matching confirm that there is no overlap between our training data (FineHARD, SAMM) and the evaluation benchmarks. The performance improvements reported in this paper are therefore attributed to the model's robust reasoning capabilities rather than memorization of testing samples.

