# OpenReview forum: "CORE: Conflict-Oriented Reasoning for General Multimodal Manipulation Detection"
_ICML.cc/2026/Conference — ICML 2026 regular_

### Official Review · Reviewer_UmZE · 2026-02-21

**Soundness:** 3
**Presentation:** 3
**Significance:** 3
**Originality:** 3
**Overall Recommendation:** 5
**Confidence:** 4

**Summary:**

This work provides fine-grained annotated multi-modal conflict attribution corpus (CAC) and a conflict-oriented reasoning fake news detection framework. The framework endows MLLMs with the human-like ability to fastly adapts to unseen misinformation patterns, moving beyond the conventional paradigm for general multimodal manipulation detection.

**Compliance With Llm Reviewing Policy:**

Affirmed.

**Final Justification:**

As I indicated in the acknowledgement, the response addressed my concerns. I keep my prior assessment.

**Key Questions For Authors:**

- What is the proportion of conflicts in the CAC dataset that are purely caused by physical manipulation traces?
- How does the proposed framework perform on the subset of the CAC dataset that contains conflicts purely caused by physical manipulation traces? This would help to understand the effectiveness of the proposed framework in handling such cases.
- How is the weight ratio between the contrastive loss and the reasoning loss determined?

**Limitations:**

yes

**Strengths And Weaknesses:**

- Soundness: The proposed framework and the motivation is well-grounded and technically sound. The paper also provides detailed analysis of the proposed framework, which is insightful and helpful for future research.
- Presentation: The paper is well-written and easy to follow. The motivation and the proposed framework are clearly described.
- Significance: The proposed framework is a novel and effective approach aiming to tackle the generalization problems in multimodal misinformation detection, especially focusing on adaptation to the non-seen manipulation patterns. The proposed CAC dataset is also a valuable resource for future research in this area.
- Originality: The paper proposes a novel framework for multimodal misinformation detection, which is a significant departure from the conventional paradigm. The proposed CAC dataset is also a novel contribution to the multimodal fake news detection field.

---

> ### Author Rebuttal · Authors · 2026-03-29
>
> # Response to Reviewer UmZE
>
> We thank the reviewer for recognizing the **soundness** of our framework and motivation, the **clear presentation** of the paper, the **significance** of CORE for generalizable multimodal misinformation detection, and the **originality** of both our conflict-oriented perspective and the CAC dataset. Regarding the concerns, we provide the following clarifications:
>
> > ### **Q1: Proportion of physical-trace conflicts in CAC**
>
> In CAC, conflicts purely caused by physical manipulation traces account for approximately **23.5\%** of the data.
>
> > ### **Q2: Performance on physical-trace manipulation cases**
>
> We agree that this is an important analysis. Since current multimodal manipulation benchmarks rarely isolate purely physical manipulation cases, we use the corresponding subset in **DGM4**, which is the benchmark in our evaluation suite that explicitly contains such visual-only manipulation patterns. Under the **100-sample few-shot** setting, $\mathbf{CORE}_{Qwen}$ achieves the best performance on this subset, outperforming both FKA-Owl Qwen2.5-VL-3B, and Qwen3-VL-235B (zero-shot). This further supports the effectiveness of CORE in handling conflicts caused purely by physical manipulation traces.
>
> |Method|ACC|
> |-|-:|
> |FKA-Owl|45.2|
> |Qwen2.5-VL-3B|45.3|
> |Qwen3-VL-235B|48.6|
> |$\mathbf{CORE}_{Qwen}$|58.2|
>
> > ### **Q3: Sensitivity to loss weighting**
>
> In the standard setting, all objectives are equally weighted. We also explored different weights in Eq. (6), i.e., $\frac{2(L_{cacl}+\alpha L_{cr})}{1+\alpha}$, with $\alpha=0.5$ and $\alpha=2$ (this weighting form targets to cover the 1:1 seting in our paper).
>
> Overall, the results (100-shot) are relatively stable, suggesting that CORE is not highly sensitive to moderate changes in the loss weight. When $\alpha=2$, the benchmark performance does not change significantly, but we observe some degradation in language generation quality: under zero-shot prompting for conflict factors and sources, the model occasionally produces slightly corrupted outputs. When $\alpha=0.5$, the performance becomes somewhat weaker. We will add this sensitivity analysis in the revision.
>
> |Setting|DGM4|MDSM|MMFakeBench|NewsCLIPpings|
> |-|-:|-:|-:|-:|
> |$\mathbf{CORE}_{Qwen}$ ($\alpha=1$)|59.7|69.0|73.5|64.3|
> |$\mathbf{CORE}_{Qwen}$ ($\alpha=0.5$)|58.6|67.8|72.9|63.7|
> |$\mathbf{CORE}_{Qwen}$ ($\alpha=2$)|60.1|69.2|73.2|64.8|

---

> > ### Author Rebuttal · Reviewer_UmZE · 2026-04-02
> >
> > Thanks, you've fully resolved my concerns about your works.

---

### Official Review · Reviewer_gvXG · 2026-03-03

**Soundness:** 3
**Presentation:** 2
**Significance:** 2
**Originality:** 3
**Overall Recommendation:** 4
**Confidence:** 2

**Summary:**

The paper proposes the CORE framework, which constructs a Conflict Attribution Corpus (CAC) with fine-grained annotations of conflict factors and their sources, and introduces a two-stage training scheme (MBPT and CPT) to equip MLLMs with explicit conflict-capturing capability. Experiments show consistent improvements under few-shot settings, cross-manipulation zero-shot evaluation, and large-scale training scenarios. The overall objective is to establish a unified detection paradigm grounded in the intrinsic notion of “conflict,” rather than designing specialized modules for particular manipulation artifacts.

**Compliance With Llm Reviewing Policy:**

Affirmed.

**Key Questions For Authors:**

Is conflict a necessary condition for detecting forgery? Are there cases where manipulated content contains no obvious conflict yet is still fake? Could the construction of CAC introduce implicit overlap with the test distribution? If the backbone were instruction-tuned with a comparable amount of data but without explicit conflict supervision, how would performance compare? In addition, does conflict-based learning remain effective under more severe domain shifts or cross-context scenarios?

**Limitations:**

The approach relies heavily on world knowledge and the capabilities of large models; if the underlying knowledge is incomplete or outdated, conflict detection performance may degrade. The construction of CAC is resource-intensive, and its scalability in practice remains to be validated. Performance slightly declines when the CPT data scale increases, suggesting potential overfitting to certain conflict patterns. Moreover, if misused, structured conflict annotations could inadvertently facilitate the generation of more subtle and less detectable manipulations.

**Strengths And Weaknesses:**

Strengths:

Treating “conflict” as a core property of multimodal forgery offers a more unified conceptual perspective than traditional approaches that target specific manipulation patterns, and it has stronger potential for generalization.

The CAC dataset provides structured supervision on conflict factors and sources, enabling the model to learn explicit conflict structures rather than relying solely on binary real/fake labels.

The experimental evaluation is relatively comprehensive, covering few-shot adaptation, cross-type zero-shot generalization, scaling analysis, and ablation studies. The reported performance gains are substantial and generally convincing.

Weaknesses:

The assumption that “conflict” constitutes the essence of forgery is not theoretically formalized. The paper does not sufficiently discuss whether all high-quality manipulations necessarily exhibit perceivable conflicts.

CAC is entirely generated and cross-validated by MLLMs, which may introduce teacher bias or distribution coupling risks. The analysis of data independence and potential leakage remains limited.

There is no thorough comparison with stronger prompting strategies or instruction tuning at a comparable data scale, making it difficult to fully rule out the possibility that the gains stem from additional supervision rather than the proposed conflict-oriented design.

---

> ### Author Rebuttal · Authors · 2026-03-29
>
> # Response to Reviewer gvXG
>
> We thank the reviewer for recognizing the unified conflict-oriented perspective of our framework, the value of CAC with structured conflict supervision, and the comprehensive empirical evaluation with strong few-shot and zero-shot generalization. Regarding the concerns, we provide the following clarifications:
>
> > ### **Q1: Conflict as an intuitive principle rather than a strict theory**
>
> Admittedly, the paper does not provide a formal theoretical proof that conflict is the essence of forgery. Our claim is instead an intuitive and operational one: humans often identify fake news by noticing conflicts between modalities, physical relations, commonsense, or world knowledge, and our results support the effectiveness of this perspective in both in-domain and OOD zero-shot settings.
>
> As for whether all high-quality manipulations exhibit perceivable conflicts, we view the answer as yes in an operational sense. Here, conflict is defined broadly as any inconsistency between manipulated content and its supporting evidence. Higher-quality manipulations may make such conflict more subtle, but do not remove it; otherwise, the content would be indistinguishable from real news for any detector.
>
> > ### **Q2: Teacher bias and data leakage**
>
> We mitigate teacher bias via a carefully designed pipeline. As detailed in Appendix I, CAC follows a retrieval–verify–generate–verify–structure–verify process. We avoid using the same MLLM for both generation and verification by employing different models for cross-stage validation, reducing bias and distribution coupling. Human verification of 1,000 random samples achieves a 99.3% acceptance rate, confirming CAC quality.
>
> For data independence, Appendix M provides a dedicated overlap analysis for MBPT, CPT, and the evaluation benchmarks, and shows no evidence of leakage. Appendix M.2 further supports this with additional experiments. Therefore, the reported gains are not explained by teacher bias or data leakage.
>
> > ### **Q3: Comparison with stronger prompting or instruction tuning**
>
> We already include closely matched comparisons for both prompting and instruction tuning. For prompting, Appendix L shows that direct prompting remains clearly below CORE. For instruction tuning, Appendix B, Tables 5 and 6 show that removing $L_{cl}$ or $L_{cacl}$ reduces training to mainly language-generation supervision in CAC, without the key objectives for modality bridging or conflict-aware separation, and performance drops accordingly. These results suggest that the gains do not come from additional supervision alone, but from the conflict-oriented design itself.
>
> > ### **Q4: Effectiveness under stronger domain/context shifts**
>
> Appendix K reports cross-dataset zero-shot results on multiple benchmarks with substantially different source distributions and manipulation pipelines (See benchmark introduction in Appendix C), showing that CORE remains effective under severe domain shifts. Appendix J further evaluates time-sensitive events beyond the model’s pretraining-time knowledge scope, where CORE also maintains strong performance. Together, these results support the effectiveness of conflict-based learning under stronger domain and context shifts.
>
> > ### **Q5: What if the underlying knowledge is outdated**
>
> We discuss this scenario in Appendix J through the **Time-Sensitive Events** evaluation, where the underlying knowledge in CAC has been outdated and our model still keeps superior performance (see Table 11).
>
> > ### **Q6: CAC scalability**
>
> We will fully release CAC for research use, so future work on CORE does not require reconstructing the dataset from scratch. In addition, Table 4(d) already shows that the CAC scale (19K) can be treated as a standard/default setting: further enlargement does not bring clear gains and may instead increase the risk of overfitting to specific manipulation patterns. Appendix J further shows that CORE with CAC-19K remains effective on time-sensitive events in evolving real-world scenarios, supporting its practical scalability. For possible expansion, we will release detailed CAC extension code and documentation.
>
> > ### **Q7: Larger CPT scale and overfitting**
>
> As you mentioned, contantly increasing CAC scale can indeed make the model fit more specific manipulation patterns. This is exactly why we study the CPT data scale in Table 4(d), which helps identify an appropriate training scale and avoid this issue. At the same time, our strong zero-shot and Time-Sensitive Events results indicate that CORE still generalizes well at the standard CAC scale (19K), so overfitting is unlikely to be a major concern in practice.
>
> > ### **Q8: Potential misuse of CAC**
>
> We agree that this risk should be taken seriously. In the release process, we will apply strict access control and review to CAC, and ensure that it is only used for multimodal manipulation detection research.

---

> > ### Author Rebuttal · Reviewer_gvXG · 2026-04-03
> >
> > My questions have been resolved, so I am keeping my score.

---

### Official Review · Reviewer_b7yC · 2026-03-11

**Soundness:** 2
**Presentation:** 3
**Significance:** 3
**Originality:** 2
**Overall Recommendation:** 2
**Confidence:** 3

**Summary:**

This paper proposes CORE, a conflict-oriented framework for general multimodal manipulation detection. Instead of relying on manipulation-specific artifacts, the method aims to detect intrinsic semantic and physical conflicts in multimodal fake news. The approach includes two stages: Modality Bridging Pre-Training (MBPT) on FineHARD and Conflict Perception Training (CPT) on a newly constructed Conflict Attribution Corpus (CAC) with 19,532 samples. The reported few-shot and zero-shot results on several benchmarks are strong.

**Compliance With Llm Reviewing Policy:**

Affirmed.

**Key Questions For Authors:**

As shown above

**Limitations:**

Yes. The paper includes a discussion of ethical risk and controlled release. However, computational cost and deployment efficiency should also be discussed more explicitly.

**Strengths And Weaknesses:**

Strengths: The paper targets an important and timely problem.

Weaknesses:
1. CAC is constructed with external retrieval and strong MLLMs, so it is hard to tell whether CORE learns conflict reasoning itself or mainly distills the priors and biases of the synthetic pipeline.
2. Since the dataset is the main source of supervision, the current level of human verification does not seem sufficient to fully establish its reliability at this scale.
3. The method relies on search results during data construction, but the paper does not examine noisy, irrelevant, or adversarial retrieval, which seems important for the intended real-world setting.
4. Most of the support comes from downstream accuracy, t-SNE plots, and feature separability. I do not find this sufficient to conclude that the model has learned better reasoning rather than better task-specific discrimination.
5. CPT pushes conflicting concepts apart in feature space, but the paper does not clearly explain why this should capture contradiction, instead of simply encouraging separability.
6. The reported gains come after substantial auxiliary supervision and pretraining, so they should not be attributed to few-shot adaptation alone.
7. Some benchmark choices and the limited discussion of overlap make it harder to judge how much of the improvement reflects true generalization to unseen manipulations.
8. The modality connector / aligner is an important part of the method, but its architecture and initialization are not described clearly enough.
9. The paper does not provide enough discussion of inference cost, latency, or deployment trade-offs relative to lighter baselines.

---

> ### Author Rebuttal · Authors · 2026-03-29
>
> # Response to Reviewer b7yC
>
> We thank the reviewer for recognizing the importance of manipulation detection and CORE’s strong few- and zero-shot performance across benchmarks. We address your concerns as follows:
>
> > ### **Q1: Synthetic supervision vs. conflict learning**
>
> This point is already reflected in Appendix B, Tables 5 and 6. Removing $L_{cl}$ or $L_{cacl}$ largely reduces training to fitting the synthetic language supervision in CAC, without the key objectives for modality bridging or conflict-aware feature separation. Under this setting, performance drops clearly, showing that CORE does not mainly gain from distilling synthetic priors, but from the conflict learning introduced by $L_{cl}$ and $L_{cacl}$. Table 2 further supports this point: even strong general-purpose MLLMs still perform unsatisfactorily on this task, suggesting that CORE’s gains are unlikely to come from merely distilling their priors.
>
> > ### **Q2: Reliability of CAC as large-scale supervision**
>
> We ensure CAC quality through both pipeline design and human verification. CAC is built with a rigorous retrieval-verify-generate-verify-structure-verify pipeline (Appendix I.1-I.3) to reduce noise in conflict factors and sources. We further conduct human verification in Appendix I.4: 5 annotators check 1,000 (about **5\%** of the dataset) randomly sampled examples, and 993 pass. These results support the reliability of CAC at scale.
>
> > ### **Q3: Robustness to retrieval noise**
>
> As described in Appendix I.1, retrieval is not used in raw form: keywords are first generated by MLLMs, the retrieved materials are then verified before use, and CAC is further filtered through a strict generate-verify pipeline (see Q2). Consistent with this, the final human check shows a 99.3\% acceptance rate on 1,000 sampled examples. This suggests that noisy or irrelevant retrieval does not materially affect the final quality of CAC.
>
> > ### **Q4: Reasoning vs. task-specific discrimination**
>
> Our evidence goes beyond in-domain accuracy, t-SNE, and feature separability. Specifically, CORE also maintains strong performance under out-of-domain settings (Tables 11, 12, and 4(a)). If it mainly learns task-specific discrimination, such gains would be much less likely to transfer consistently across domains. In addition, Figure 2 shows that CORE improves conceptual discriminability, which is the basis for capturing conflicting cues in fake news. Together, these results support that CORE improves conflict perception and reasoning ability, rather than only learning a stronger task-specific decision boundary.
>
> > ### **Q5: Why concept distancing captures contradiction**
>
> Pushing conflicting concepts apart in feature space is indeed a form of encouraging separability. In this task, contradiction is precisely reflected as separability between semantically incompatible concepts. If such concepts remain entangled in feature space, the model cannot reliably perceive their inconsistency. By explicitly pushing them apart, CORE reshapes the representation space so that incompatible concepts become easier to distinguish, and their conflict becomes a direct discriminative signal for fake news detection. Separability is the mechanism by which contradiction is encoded and recognized.
>
> > ### **Q6: Few-shot gains vs. auxiliary supervision**
>
> We agree that the gains should not be attributed to few-shot adaptation alone. They mainly come from the auxiliary supervision and pre-training in CORE, which is the core contribution of this work. The few-shot stage only serves as a downstream validation setting, and all methods are evaluated under the same few-shot protocol.
>
> > ### **Q7: Generalization beyond overlap**
>
> We have actually examined the overlap issue in Appendix M.  Appendix M discusses the potential overlap among the data used in MBPT, CPT, and evaluation benchmarks, and the analysis in M.2 shows that no overlap is observed in the used data. Therefore, the reported gains are not explained by train–test contamination, but reflect genuine generalization to unseen manipulations.
>
> > ### **Q8: Aligner implementation details**
>
> As stated in Line 242 Page 5, the aligner is implemented as a **cross-attention layer**. For initialization, we do not use any special design; it follows standard **Kaiming initialization**. The modality connector follows the backbone design; for example, in Qwen2.5VL it's the MLP projector.
>
> > ### **Q9: Inference cost and deployment trade-offs**
>
> As stated in Sec. 4.4, CORE directly predicts Fake/Real at inference time, without generating extra conflict factors or sources, and the Aligner is discarded after training. Therefore, inference is essentially the same as a standard MLLM forward pass, with cost and latency comparable to the backbone (e.g., Qwen2.5-VL-3B). We will clarify this more explicitly in the revision.
> |Method|Params|Latency(ms)|ACC|
> |-|-:|-:|-:|
> |Qwen2.5-VL-3B|3B|390|52.1|
> |RamDG|0.2B|12.3|52.9|
> |$\mathbf{CORE}_{Qwen}$|3B|390|66.6|

---

> > ### Author Rebuttal · Reviewer_b7yC · 2026-04-04
> >
> > After reviewing carefully, I keep this score.

---

### Official Review · Reviewer_eMWy · 2026-03-13

**Soundness:** 2
**Presentation:** 2
**Significance:** 3
**Originality:** 3
**Overall Recommendation:** 4
**Confidence:** 3

**Summary:**

This paper proposes CORE (Conflict-Oriented Reasoning), a multimodal manipulation detection framework that explicitly models conflicts arising from cross-modal inconsistencies and violations of world knowledge. To this end, the paper introduces the Conflict Attribution Corpus (CAC), which provides fine-grained annotations of conflict factors and their sources. The authors evaluate CORE on multiple multimodal manipulation benchmarks (e.g., DGM4, MMFakeBench, and NewsCLIPpings) and demonstrate that it consistently outperforms prior methods, particularly in few-shot and zero-shot settings.

**Compliance With Llm Reviewing Policy:**

Affirmed.

**Final Justification:**

Based on the author's feedback and considering other reviewers' opinions, I improved my score accordingly. However, I am still confused about this work, given that the proposed method implicitly supervises the reasoning process by guiding the model to discover it. A more reasonable approach is to fine-tune models using the chains rather than the weird methods proposed in this work, despite their effectiveness.  Constraining the model's reasoning process in this way is not necessarily generalizable or appropriate in certain cases.

**Key Questions For Authors:**

NAN

**Limitations:**

Not

**Strengths And Weaknesses:**

## Strengths

- **Intuitive motivation.** Modeling manipulated misinformation as a combination of semantic distortion and commonsense conflict is well-justified.
- **Thorough evaluation.** Experiments are comprehensive, spanning both few-shot and zero-shot settings.
- **Valuable dataset contribution.** The CAC dataset provides fine-grained annotations of conflict factors and sources.

## Weaknesses

- **Key baselines are missing.** The study should include stronger comparisons, such as: (i) directly prompting the MLLM/MLVM without additional training using an analogous prompting pipeline; (ii) LLM-generated $C_1/C_2$ followed by LLM-based detection; and (iii) an instruction-tuned MLLM evaluated without MBPT/CPT.
- **Insufficient implementation and training details.** Critical choices are underspecified, including the backbone models and hyperparameters for the modality connector, cross-modal aligner, and LLM. Figure 4 suggests the LLM is trained, but Section 4.3 does not clearly indicate whether a next-token prediction objective is used to preserve language modeling capability; moreover, Eq. (5) appears misaligned with the standard LLM training loss. The inference procedure is also unclear—specifically, whether conflict factors $C_1, C_2$ and sources $S_1, S_2$ are generated at test time and how they are used.
- **Limited scope of applicability.** CORE appears primarily tailored to misinformation detectable via factual inconsistency, and it does not use external retrieval at inference. This likely limits performance on purely visual manipulations that do not contradict text or world knowledge, as well as time-sensitive events that fall outside the model’s pretraining knowledge

---

> ### Author Rebuttal · Authors · 2026-03-29
>
> # Response to Reviewer eMWy
>
> We thank the reviewer for the positive assessment of our work, especially for recognizing **the intuitive motivation** of modeling multimodal misinformation through conflict-oriented reasoning, the **thorough evaluation** across few-shot and zero-shot settings, and the value of our **CAC dataset with fine-grained conflict annotations**. Regarding concerns, we provide clarifications:
>
> > ### **Q1: Regarding the key baselines**
>
> We would like to clarify that these comparisons are already provided in our manuscript.
>
> (i) Directly prompting the MLLM. This comparison is already included in Appendix L. The results show that prompting alone is clearly insufficient: compared with prompt-only Qwen2.5-VL-3B, $\mathbf{CORE}_\mathbf{{Qwen}}$-3B achieves an average gain of **18.2%** across  the four benchmarks. This shows that the improvement does not come from prompting alone, but from CORE training.
>
> (ii) LLM-generated reasoning followed by LLM-based detection. We understand this as asking whether language-level reasoning supervision alone could explain the gain. This setting is closely approximated by Appendix B, Table 6, w/o $L_{cacl}$. In this variant, the conflict-aware contrastive objective is removed, leaving only language-level reasoning supervision. Compared with the full CORE setting, performance drops by **11.75% on average** over the four benchmarks. This indicates that reasoning generation alone is insufficient, and explicit conflict-aware learning is necessary.
>
> (iii) An instruction-tuned evaluated without MBPT/CPT. This setting is already reflected in Tables 5 and 6. Specifically, removing $L_{cl}$ makes MBPT degenerate into only the object-occurrence VQA generation objective, and removing $L_{cacl}$ makes CPT degenerate into only the rationale generation objective $L_{cr}$. In both cases, training is reduced to mainly instruction-style language supervision, without the key objectives for modality bridging and conflict discrimination. Correspondingly, performance drops consistently when either objective is removed. We also compare with strong instruction-tuned MLLMs such as Qwen2.5VL-3B and Gemma3-4B in Table 2. These results show that generic instruction tuning alone is insufficient, and that CORE’s gains come from the MBPT/CPT design.
>
> To address potential oversights due to the appendix length, we will add a clear outline and per-section summaries in the final Appendix to ensure all results are easily traceable.
>
> > ### **Q2: Regarding insufficient implementation and training details**
>
> (i) Our backbones have been provided in Line 359 page 7 including Qwen2.5VL-3B and Gemma3-4B. Our method actually does not introduces important hyperparameters, all implementaiton details are provided in section 5 and Appendix.A.  The architecture of the cross-modal aligner, a **single-layer cross-attention module**, is specified in Line 22 Page 5. The modality connector follows the backbone design; for example, in Qwen2.5VL it's the **MLP projector**.
>
> (ii) Our method trains the LLM, and the $L_{o2vqa}$ in MBPT and $L_{cr}$ are formuated as the language modeling loss (next token prediction)， which is explained in Line 271 Page 5 and Line 302 Page 6. For Eq.(5), it's indeed not a LLM training loss, but our conflict-aware contrastive loss, aiming to reshape the feature space and improve separability between conflicting concepts.
>
> (iii) As stated on Line 327 Page 6, at test time, the model **directly outputs the final answer**, and does **not** need to explicitly generate $S_1, S_2, C_1,$ or $C_2$.
>
> We will release codes and checkpoints with clear guidance to clarify all details of our architecture and reproduce our results.
>
> > ### **Q3: Regarding the limited scope of applicability**
>
> Our method is not restricted to manipulations detectable only through factual inconsistency with static world knowledge.
>
> (i) **Purely visual manipulations.** Actually, such cases are categorized as image-image conflicts (see Fig. 9, second row, second column) in CAC's annotation to handle purely visual manipulations. Moreover, evaluation benchmarks such as MMFakeBench and MDSM include samples where forgery cues are primarily visual; for example, MDSM contains manipulated images paired with semantically aligned text. The strong performance on these benchmarks supports CORE’s applicability beyond factual inconsistency with world knowledge.
>
> (ii) **Time-sensitive events.** We have already included dedicated experiments in Appendix J. The results show that CORE remains effective on events beyond the temporal scope of the model’s pretraining knowledge, suggesting that it does not rely only on memorized knowledge but also benefits from learned conflict-oriented reasoning.
>
> We agree that the lack of external retrieval at inference may limit performance on rapidly evolving real-world events and will explore integrating retrieval as a tool-use module to enhance real-time detection in future work.

---

> > ### Author Rebuttal · Reviewer_eMWy · 2026-04-03
> >
> > I find "Similarly, during inference, the model requires only this simple instruction to
> > make predictions." in  Line 327 Page 6,  but did not find  "the model directly outputs the final answer, and does not need to explicitly generate".
> >
> > Remain concerns:
> >
> > The proposed method suffers from limitations in its ability to produce this reasoning process. Why does your method not produce this content?

---

> > > ### Author Response · Authors · 2026-04-04
> > >
> > > We thank the reviewer for this follow-up and respectively response your concerns as following:
> > >
> > > > ### **Clarification of the inference setting**
> > >
> > > In the last round response, we regret not giving a clear explanation for the statement in our paper: "Similarly, during inference, the model requires only this simple instruction to make predictions." In this claim, "this simple instruction" refers to "Is the news real or fake?" , and the model directly generates the answer "real or fake" using this simple prompt. Our explanation in the last round："the model directly outputs the final answer, and does not need to explicitly generate ..." — attempted to clarify this, but seems to have caused confusion. We apologize for this confusion.
> > >
> > > In our revision, we would add a sufficiently clear and detailed explanation for the model inference.
> > >
> > > > ### **Regarding your concern of "The proposed method suffers from limitations in its ability to produce this reasoning process."**
> > >
> > > We would clairfy this concern by answering following questions:
> > >
> > > - ### **Is our CORE able to produce reasons?**
> > >
> > > **Yes.** During CPT, CORE is explicitly trained with the instruction **“Does the news Real or Fake? If it’s fake, further give the reason.”**, with the target format **“Real. / Fake. Because the C1 from S1 conflicts with C2 from S2.”** Therefore, when we prompt the model with this prompt, CORE can generate a **conflict-based reasoning process** in addition to the final prediction during inference.
> > >
> > > - ### **What's the quality of the generated reasons then?**
> > >
> > > To further address the reviewer’s concern, we add an additional evaluation dedicated to reasoning generation.
> > >
> > > Specifically, we build a new evaluation subset by sampling **1k instances** from each of **DGM4, MDSM, MMFakeBench, and NewsCLIPpings**, and annotate conflict-based rationales with the same rigorous pipeline used in CAC construction, which produces **486** candidate samples. We then **manually check every sample**, and retain **483** valid samples with reliable rationale annotations for the final evaluation. On this set, we evaluate two settings: **zero-shot** and **100-shot**. The results are as follows:
> > >
> > > | Setting | Veracity ACC | Reasoning quality / rationale match |
> > > |---|---:|---:|
> > > | Zero-shot | 60.4 | 96.0 |
> > > | 100-shot | 67.1 | 97.2 |
> > >
> > > Here, **Reasoning quality / rationale match** measures whether the generated reason is **semantically consistent** with the annotated rationale for a **correctly predicted sample**. To evaluate this, we first use **one LLM as the judge** to compare the generated reason with the annotated rationale, and then use **another LLM for cross-validation** to further ensure the reliability of the judgment.
> > >
> > > - ### **Why we did not report the reasoning results in our paper?**
> > >
> > > The primary reason is that the lack of conflict-oriented reason annotation in existing benchmarks for reliable evaluation. Existing multimodal manipulation benchmarks only provide **veracity labels**, but do not provide **standardized rationale annotations** for evaluating whether a generated reason is correct. This is exactly why we construct **CAC**: to provide fine-grained conflict supervision for training.
> > >
> > > We agree generating reasons for results explanation is significant and meaningful and will explore this research direction in our future work.
> > >
> > > -  ### **Takeaway**
> > >
> > > These results show that CORE is able not only to predict the final label, but also to generate **structured conflict-based reasoning** when explicitly prompted to do so. We will add this clarification and the new reasoning-generation evaluation in the revised version:
> > >
> > > ***During inference, we simply prompt the model with instruction "Is the news real or fake?" to produce the "real/fake" veracity prediction. Note that our inference stage does produce conflict-related reason due to the lack of corresponding annotaitons in existing benchmarks.***

---

### Decision · Program_Chairs · 2026-04-30

**Decision:**

Accept (regular)

**Comment:**

The paper introduces CORE, a framework that detects multimodal fake news by modeling intrinsic conflicts between modalities or with world knowledge.
A key contribution is the Conflict Attribution Corpus (CAC), which provides fine-grained annotations to supervise this reasoning process.

In the rebuttal and discussion period, reviewers generally praised the intuitive motivation, the value of the new dataset, and the strong empirical performance, particularly in zero-shot and few-shot settings. While some concerns were raised regarding whether the model learns true reasoning versus task-specific discrimination or if the synthetic data was reliable, the authors provided convincing evidence through ablation studies, out-of-domain evaluations, and a rigorous multi-stage verification pipeline for the CAC dataset. The authors also clarified that while the model is optimized for veracity prediction, it remains capable of generating structured rationales when prompted.

The consensus among the reviewers who engaged with the rebuttal is that the technical contributions are solid and the results demonstrate genuine generalization to unseen manipulations. Therefore, I recommend this paper for weak accept.